# A fast-acting lipid checkpoint in G1 prevents mitotic defects

Marielle S. Köberlin [1,2] ✉, Yilin Fan[1,6], Chad Liu [1,7], Mingyu Chung[1,8], Antonio F. M. Pinto[3], Peter K. Jackson[2,4], Alan Saghatelian[3] & Tobias Meyer [1,5] ✉

Lipid synthesis increases during the cell cycle to ensure sufficient membrane mass, but how insufficient synthesis restricts cell-cycle entry is not understood. Here, we identify a lipid checkpoint in G1 phase of the mammalian cell cycle by using live single-cell imaging, lipidome, and transcriptome analysis of a non-transformed cell. We show that synthesis of fatty acids in G1 not only increases lipid mass but extensively shifts the lipid composition to unsaturated phospholipids and neutral lipids. Strikingly, acute lowering of lipid synthesis rapidly activates the PERK/ATF4 endoplasmic reticulum (ER) stress pathway that blocks cell-cycle entry by increasing p21 levels, decreasing Cyclin D levels, and suppressing Retinoblastoma protein phosphorylation. Together, our study identifies a rapid anticipatory ER lipid checkpoint in G1 that prevents cells from starting the cell cycle as long as lipid synthesis is low, thereby preventing mitotic defects, which are triggered by low lipid synthesis much later in mitosis.

The decision to divide or not is a fundamental biological problem. Although cells require nutrients and lipids for a successful cell division[1], the role of lipids in the cell-cycle decision process is unknown. Traditionally, cell-cycle decision research has focused primarily on the role of mitogens and DNA damage[2-4].

To ensure the faithful completion of the cell cycle, signaling checkpoints are pausing sequential steps in the cell cycle until the previous step is completed or stress is resolved. Checkpoints are critical for the accurate duplication of DNA in S phase and separation of chromosomes into two daughter cells during mitosis[5]. A DNA damage checkpoint in G1 phase induces p53 to inhibit the cyclin and cyclin-dependent kinase (CDK)-mediated phosphorylation and inactivation of Retinoblastoma (Rb) protein, which prevents the duplication of damaged DNA[3,4,6-10]. However, cells must not only duplicate their DNA, but also their lipid mass during each cell cycle and also adapt the lipid composition for a successful cytokinesis and mitosis[1,11-16]. Our study set

out to determine if, similar to the DNA damage checkpoint, there is a lipid checkpoint that senses whether sufficient lipids are available for cells to progress through the cell cycle and when during the cell cycle such a lipid checkpoint engages. Regulatory signaling mechanisms are considered cell-cycle checkpoints if they (1) stop cell-cycle progression, (2) are rapidly induced, and (3) are reversed once optimal conditions are reached or the stress is resolved[17,18].

Previous studies provided contradictory evidence whether a lipid checkpoint exists. On one hand, it has long been known that cells require lipid synthesis to divide[1], suggesting a lipid checkpoint does exist. On the other hand, low lipid synthesis and deregulation of the lipid repertoire can end in cell division failure and death[11,19], suggesting there is no protective checkpoint. Furthermore, several cancers require fatty acid synthase (FASN) activity to survive, suggesting that the response to low lipid synthesis is cell death rather than lipid checkpoint regulation[20-24]. FASN synthesizes saturated fatty acids such

[1]Department of Chemical and Systems Biology, Stanford University School of Medicine, Stanford, CA 94305, USA. [2]Baxter Laboratory, Department of Microbiology & Immunology, Stanford University School of Medicine, Stanford, CA 94305, USA. [3]Clayton Foundation Laboratories for Peptide Biology and Mass Spectrometry Core, Salk Institute for Biological Studies, La Jolla, CA 92037, USA. [4]Department of Pathology, Stanford University School of Medicine, Stanford, CA 94305, USA. [5]Department of Cell and Developmental Biology, Weill Cornell Medicine, New York, NY 10065, USA. [6]Present address: Department of Pathology and Center for Cancer Research, Massachusetts General Hospital and Harvard Medical School, Boston, MA 02114, USA. [7]Present address: Chan Zuckerberg Biohub, San Francisco, CA 94111, USA. [8]Present address: Department of Neurology and Neurological Sciences, Stanford University School of Medicine, Stanford, CA 94305, USA. ✉e-mail: mkoeberlin@stanford.edu; tom4003@med.cornell.edu

as Palmitate, which can be desaturated and together are the building blocks of complex lipids including glycerophospholipids, sphingolipids, and neutral lipids[25]. Given these findings, the question remains whether a lipid cell-cycle checkpoint exists, and if cells monitor lipid synthesis to stop cell-cycle progression when lipid synthesis is low.

To examine when and how cells sense lipid synthesis, we combined high-throughput live cell microscopy with lipidome and transcriptome analyses in cells with acutely blocked de novo fatty acid synthesis. Strikingly, lipid synthesis inhibition in G1 rapidly remodeled a circular lipid coregulation network and prevented cells from entering S phase without causing cell death. A combined lipidome and transcriptome analysis revealed that the extensive changes in endoplasmic reticulum (ER) lipid composition caused by low lipid synthesis activate the PERK/ATF4-mediated ER stress pathway. Quantitative imaged-based single-cell analysis showed that the lipid-induced response triggers a G1 arrest that is independent of p53 but instead inhibits S phase entry by increasing the CDK inhibitor p21 and reducing Cyclin D protein levels. This in turn suppresses Rb phosphorylation and E2F activation. Markedly, the lipid-induced suppression of cell-cycle entry is acute and reversible, meeting the essential criteria for a G1 checkpoint response. In contrast, inhibition of lipid synthesis later in the cell cycle triggered mitotic defects, explaining why FASN inhibitors can also cause a growth disadvantage and are in clinical trials for the treatment of several cancers[26]. Thus, to prevent fatty acid insufficiency from triggering mitotic defects at the end of the cell cycle, cells have an anticipatory lipid checkpoint in G1 that inhibits cell-cycle entry. Finally, our study introduces a powerful strategy to combine lipidome and transcriptome analyses to identify lipid signaling mechanisms.

## Results
### Inhibition of fatty acid metabolism in G1 rapidly inhibits S phase entry
To learn whether there is a lipid checkpoint and why FASN inhibitors regulate the cell cycle and cause growth disadvantages, we used a non-transformed cellular model that does not require extracellular lipid uptake to enter the cell cycle. We synchronized human MCF-10A breast epithelial cells by growth medium removal (starvation) and added epidermal growth factor (EGF) to induce cell-cycle entry (hereafter EGF release) (Fig. 1A). We tested the importance of fatty acid synthesis by blocking different catalytic active sites of the enzyme FASN with three well-characterized chemical inhibitors C75[27], GSK2194069 (hereafter GSKi)[28], and Cerulenin[29]. We measured the percentage of cells that reached S phase using automated quantitative imaged-based cytometry (QIBC)[30,31] to monitor DNA incorporation of 5-ethnyl-2′-deoxyuridine (EdU) during replication after a 15-minute pulse.

After 20 hours of EGF release, approximately 25% of control cells have reached S phase. Using the three FASN inhibitors to block FASN activity in the EGF released cells strongly inhibited S phase entry in a dose-dependent manner (Fig. 1B, C, and S1A–C). Upstream of FASN, ATP-citrate synthase (ACLY) and Acetyl-CoA carboxylase (ACACA) catalyze the production of Acetyl- and Malonyl-CoA[32,33]. Inhibition of these enzymes using the chemical inhibitors SB204990[34] or TOFA[35] respectively, also dose-dependently reduced S phase entry after 20 hours without reducing cell numbers (Fig. 1D and S1D). Phospholipases also supply free fatty acids by catalyzing the cleavage of existing pools of glycerophospholipids[36]. Inhibition of the phospholipase iPLA2 using Bromoenol Lactone (BEL)[37] resulted in a reduction of S phase entry as well (Fig. 1D and S1D).

Downstream of free saturated fatty acids, fatty acid ligases (ACSL) and stearoyl-CoA desaturase (SCD) are important for activating[38] and desaturating fatty acids[39], respectively. Inhibition of ACSL using Triacsin C[40] or SCD using CAY10566[41] (hereafter SCDi) also resulted in a reduction in S phase entry after 20 hours without impacting cell numbers (Fig. 1D and S1D). In contrast, inhibition of either fatty acid

oxidation catalyzed by carnitine palmitoyltransferase (CPT1A) using Etomoxir[42] or triglyceride synthesis catalyzed by DGAT1 using T-863[43] did not result in a reduction of S phase entry after 20 hours (Fig. 1D and S1D). These results indicate that replenishing the free fatty acid pool is generally important to drive S phase entry.

Even a three-hour acute inhibition of FASN activity at nine hours after EGF release, resulted in significantly decreased S phase entry, indicating that fatty acid synthesis is sensed by cells later in G1 (Fig. S1E). Genetic perturbation of FASN using siRNA or CRISPR/Cas9-mediated suppression in a heterogenous cell population each significantly reduced the percentage of S phase cells (Fig. S1F–I, Supplementary Tables 1 and 2). The less pronounced decrease in S phase entry, compared to the chemical perturbation, is likely due to a relatively small reduction achievable by siRNA knockdown or CRISPR targeting (Fig. S1G, I).

When treated with C75, we observed a similar inhibition of S phase entry in hTERT-immortalized retinal pigment epithelial cells (RPE1), also without reducing cell numbers (Fig. S1J), suggesting that different human cell types are sensitive to FASN inhibition. When cells lose FASN activity they rely strongly on other sources such as lipid droplets or lipid uptake[44,45]. To test if extracellular lipids were able to compensate for the inhibition of FASN, we added back media containing serum lipids, which fully rescued S phase entry in the presence of C75 (Fig. 1E).

To understand the temporal dynamics of how FASN inhibition impacts cycling cells, we used live imaging and automated tracking of a dual fluorescent reporter system to measure CDK (Cyclin E/Cyclin A cyclin-dependent kinase) activity[46,47] and the start of S phase using a fluorescent reporter for E3 ubiquitin ligase anaphase-promoting complex/cyclosome-Cdh1 (APC/C$^{Cdh1}$) inactivation[3,48] (Fig. 1F). We imaged thousands of asynchronously cycling MCF-10A cells that were treated with different FASN inhibitors. After automated nuclear segmenting and individual cell tracking, we then computationally gated for populations of cells that received the inhibitors during different phases of the cell cycle (Fig. 1F). The population of cells that was in G0 or G1 (CDK activity = 0.4–0.6) when treated with FASN inhibitors showed strongly reduced numbers of cells increasing their CDK reporter activity or inactivating APC/C$^{Cdh1}$, indicating that only a few cells entered S phase as compared to the DMSO control (Fig. 1G, H, and S2A). The population that was in S phase (CDK activity = 1–1.2) when treated with FASN inhibitor showed no reduction in cells with high CDK activity as compared to cells treated with DNA damage inducing agent Methyl Methanesulfonate (MMS) (Fig. S2B). Thus, cells are monitoring and reacting to whether they have sufficient lipids primarily during G1.

DNA damage is a predominant cause of G1 arrest[49,50]. To determine if inhibition of FASN activity causes DNA damage, we used QIBC and computational gating in an asynchronously cycling MCF-10A cell population. FASN inhibitor treatment in S phase did not show an increase in DNA damage measured by phosphorylated histone H2A.X (p-S139) immunofluorescence (IF) (Fig. S2C) as compared to control treatments with DNA damage inducing agents MMS or neocarzinostatin (NCS). Cell death was also not increased in cells treated with different FASN inhibitors at the concentrations used to inhibit S phase entry as measured by cleaved Caspase 3 IF (Fig. S2D).

We next characterized the importance of FASN activity in later stages of the cell cycle, specifically during G2 and M phase. While almost 90% of cells treated with DMSO in late G2 underwent mitosis, only 14% of cells that received C75 earlier underwent mitosis (Fig. 1I). Around 80% of these 14% showed defects, hence were not able to successfully complete mitosis and produce two viable daughter cells (Fig. 1J, S2E). The observed mitotic defects were dose-dependent and ranged from nuclear disintegration to separation failures (Fig. 1J, S2E, F). Daughter cells that successfully completed mitosis after treatment with FASN inhibitors arrested dose-dependently in the following G1 (Fig. S2G, H).

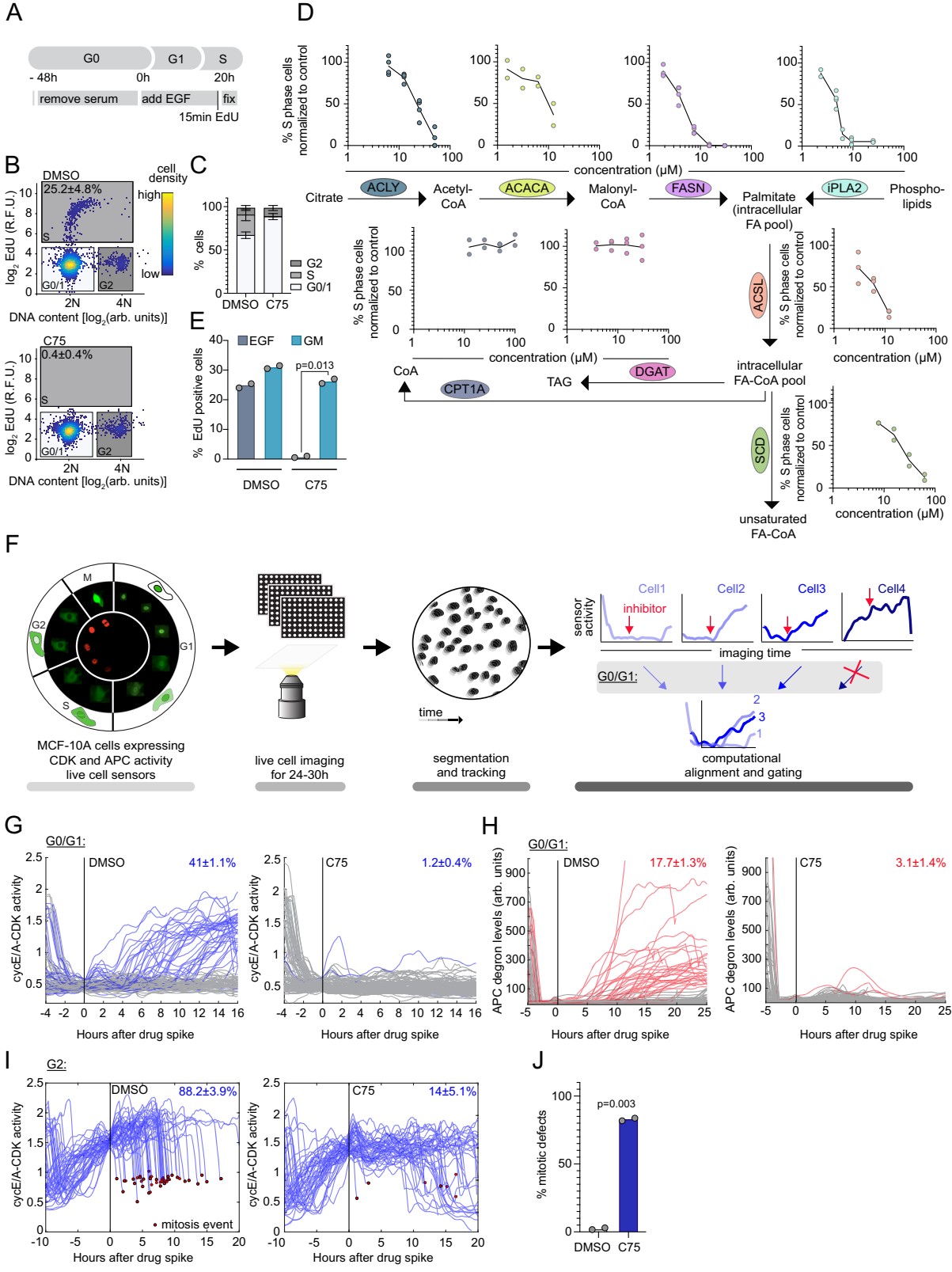

Thus, inhibition of FASN in G0 or G1 suppresses CDK activation rapidly and prevents cells from entering S phase, fulfilling the first two of the three criteria of a lipid checkpoint in G0/G1. The checkpoint caused by the loss of de novo fatty acid synthesis can also be rescued by providing serum lipids to cells, arguing that all three criteria are fulfilled. Markedly, acute FASN inhibition does not cause DNA damage or cell cycle arrest during S phase, but cells have a second time window of partial lipid sensitivity in early G2. In contrast, FASN inhibition in late G2 does not stop mitotic progression, and cells with inhibited FASN frequently show mitotic defects, arguing that lipid synthesis is functionally required in mitosis to make two daughter cells. Thus, the G1 lipid checkpoint functions as an anticipatory protective mechanism that blocks entry into S phase to prevent malfunctions occurring hours later when cells require lipid synthesis to complete mitosis.

**Fig. 1 | Inhibition of fatty acid metabolism rapidly inhibits S phase entry. A** Diagram depicting cell-cycle phases and work flow. **B** EdU signal and DNA content in EGF-released MCF-10A cells, treated with DMSO or C75 (15 μM), 3000 cells displayed. Colormap shows cell density. Data are representative of three independent experiments. Boxes show G0/1, S and G2 phase cells. Mean of high EdU percentage ± SD of three independent experiments shown. **C** Mean percentage ± SD of G0/1, S, G2 phase cells as indicated in (**B**). Data are from three independent experiments. **B**, **C** n > 17,000 cells per condition. **D** Percentage of EdU positive cells treated with increasing concentrations of fatty acid metabolism inhibitors and normalized to DMSO treatment. Metabolic pathway and targeted enzymes are shown. Data are from at least two independent experiments, n > 12,000 cells per condition. **E** Percentage of EdU positive cells treated with DMSO or C75 (15 μM) in the presence (GM: growth media) or absence (EGF) of 5% serum. Data are from two independent experiments, n > 15,000 per condition. **B**–**E** Cells were EGF-released for 20 h. **F** Schema of live imaging of MCF-10A cells expressing CDK activity (green) and APC/C$^{Cdh1}$ degron reporter (red) followed by automated nuclear segmentation,

tracking, and analysis. **G**–**I** Single-cell traces of cycling MCF-10A cells treated with DMSO or C75 (30 μM) quantifying (**G**) cyclin-E/A CDK activity after treatment during G0/G1 phase (cyclin-E/A CDK activity: 0.4–0.6): Gray traces (future cyclin-E/A CDK activity <1), blue traces (> 1). Quantification is percentage of cells with increased cyclin-E/A CDK activity, (**H**) APC/C$^{Cdh1}$ degron reporter binned by activity at point of treatment: Red traces (future degron activity >100), gray traces (<100). Quantification is percentage of cells with increased APC degron levels, (**I**) cyclin-E/A CDK activity after treatment in G2 phase (cyclin-E/A CDK activity: 1.4–1.6). Quantification is percentage of mitotic events (red dot). **G**–**I** Traces are 100 random cells. Mean percentage ± SD of two independent experiments, n > 75,000 cells per condition. **J** Percentage of mitotic defects among all mitotic events in a cell population treated with DMSO or C75 (30 μM). Mitotic defects called using H2B-mTurquoise signal. At least 159 mitotic events (average: 192) quantified per condition. Data are from two independent experiments, n > 8,000 per condition. **E**, **J** P values calculated using two-tailed paired t test. R.F.U., relative fluorescence unit. Source data are provided as Source data file.

## The lipidome is reshaped before S phase begins

To gain insights into how and when lipid changes in G1 control the lipid checkpoint, we analyzed the lipid composition as cells progress through G1 and enter S phase (Fig. 2A). S phase entry in synchronized MCF-10A cells is variable between single cells but starts between 12 and 15 hours post EGF release in many cells, as evidenced by incorporation of EdU (Fig. S3A). Another marker for G1 progression is the phosphorylation of Retinoblastoma protein (Rb)[46,51,52]. Immunofluorescence analysis shows that up to 60% of cells are Rb (p-S807/811) positive between 15 and 20 hours after EGF release indicating that most cells are in S phase at that point (Fig. S3B).

We utilized mass spectrometry-based lipidomics to analyze changes in the relative intracellular lipid abundance between starved, quiescent cells and those progressing through G1 (Fig. 2B). Markedly, rather than increasing all lipid species in G1, as one may expect from the need to double lipid mass at the time of cell division, there were strong, specific relative changes in lipid abundance before cells enter S phase (Fig. 2A, B and Supplementary Data 1). By clustering the lipid changes we identified 4 groups of lipids.

Group 1 contained lipids that decreased with cell cycle progression and was 2- to 4-fold enriched for saturated Triglycerides (TG), Monoglycerides/Diglycerides (MG/DG), Cholesterol/Cholesterol esters (Cho/ChE), and Phosphatidylserines (PS) (Fig. 2B, group 1, far left, and S3C, left).

Group 2 contained lipids that increased and was enriched up to 3-fold for glycerophospholipids including long-chained Phosphatidylethanolamines (PE), Phosphatidylinositols (PI), and most Phosphatidic acids (PA), as well as polyunsaturated TG (Fig. 2B, group 2, right, and S3C, right). PE lipids are important for membrane biosynthesis, and PI and PA mediate lipid signaling[53–55].

Group 3 contained lipids that did not change (Fig. 2B, center, and S3D), while Group 4 contained lipids that increased later in G1 and were enriched more than 7-fold for most unsaturated and some saturated free fatty acids (Fig. 2B, group 4, left, and S3C, center, S3E). These late increased lipids are likely not critical in G1 but become important later in the cell cycle.

The decreasing lipids in group 1 were significantly enriched for saturated lipids while the increasing lipids in group 2 were enriched for unsaturated lipids, and lipids in group 3 were not enriched for saturation levels (Fig. 2C). Together, our analysis identifies a decrease of saturated neutral lipids, as well as PS, and Cho/ChE. At the same time, there is an upregulation of unsaturated complex ER lipids PE, PI, PA, and TG. These lipids are important for membrane biosynthesis, fluidity, lipid signaling, and show the kinetic profile expected for lipid regulators needed to overcome a G1 lipid checkpoint response (Fig. 2D).

## Acute inhibition of FASN induces global lipid changes

The unexpected changes in lipid abundance and saturation just before S phase suggest that this is a pivotal time-period for lipid metabolism. We considered that the combination of acute lipid flux perturbations in this critical G1 phase with monitoring the resulting lipid changes would allow us to understand both the metabolic flux and checkpoint. We thus analyzed changes in lipid composition relative to control cells after acutely activating the G1 lipid checkpoint by treating EGF-released MCF-10A cells after nine hours with either of two FASN inhibitors (C75 or GSKi) or SCD inhibitor for three or six hours (Fig. 2E). We used SCD inhibition as an alternative way to perturb fatty acid metabolism and increase the relative levels of fatty acid saturation.

Treatment with both FASN inhibitors at nine hours after EGF stimulation led to an acute decrease of free fatty acid species after three hours, and further after six hours (Fig. 2F, S3F, and Supplementary Data 1). Interestingly, even complex lipids changed as early as three hours after drug addition, illustrating the rapid lipid flux from inhibiting FASN activity to altering membrane lipid composition. These lipid changes included decreased levels of many PA, all saturated MG/DG, and short-chained TG species as well as increases in unsaturated medium-chained DG and TG species (Fig. 2F and S3F, left). PA, MG/DG, and TG have key functions particularly in the ER, suggesting that changes in ER lipid composition may control the G1 lipid checkpoint.

The analysis of the relative saturation across all measured lipid species showed that FASN inhibition in cells progressing through G1 increased the overall level of unsaturated over saturated lipids (Fig. 2G), possibly as a protective response due to the reduced lipid synthesis. Notably, overall lipid changes observed after SCD inhibition were less pronounced (Fig. S3G).

Together, our data show that, rather than simply increasing lipid levels, cells dynamically remodel their lipid landscape when progressing through G1 before entering S phase. Prominent ER lipids are rapidly increased in early G1, while free fatty acid levels are increased later in G1. There is a general shift from saturated to unsaturated lipids. Furthermore, acute FASN inhibition leads to strong and rapid changes in intracellular lipid abundances and saturation levels across fatty acids, PA, DG, and TG many of which are enriched in the ER. These findings motivated us to find a way to link the global changes of the lipid landscape at this pivotal time in the cell cycle to the unknown molecular mechanism that triggers the observed G1 lipid checkpoint response.

## Rapid and global coregulation of precursor and complex lipids in G1

The lipid changes we identified at the G1 lipid checkpoint occur within an interconnected metabolic network fueled by new fatty acids. To

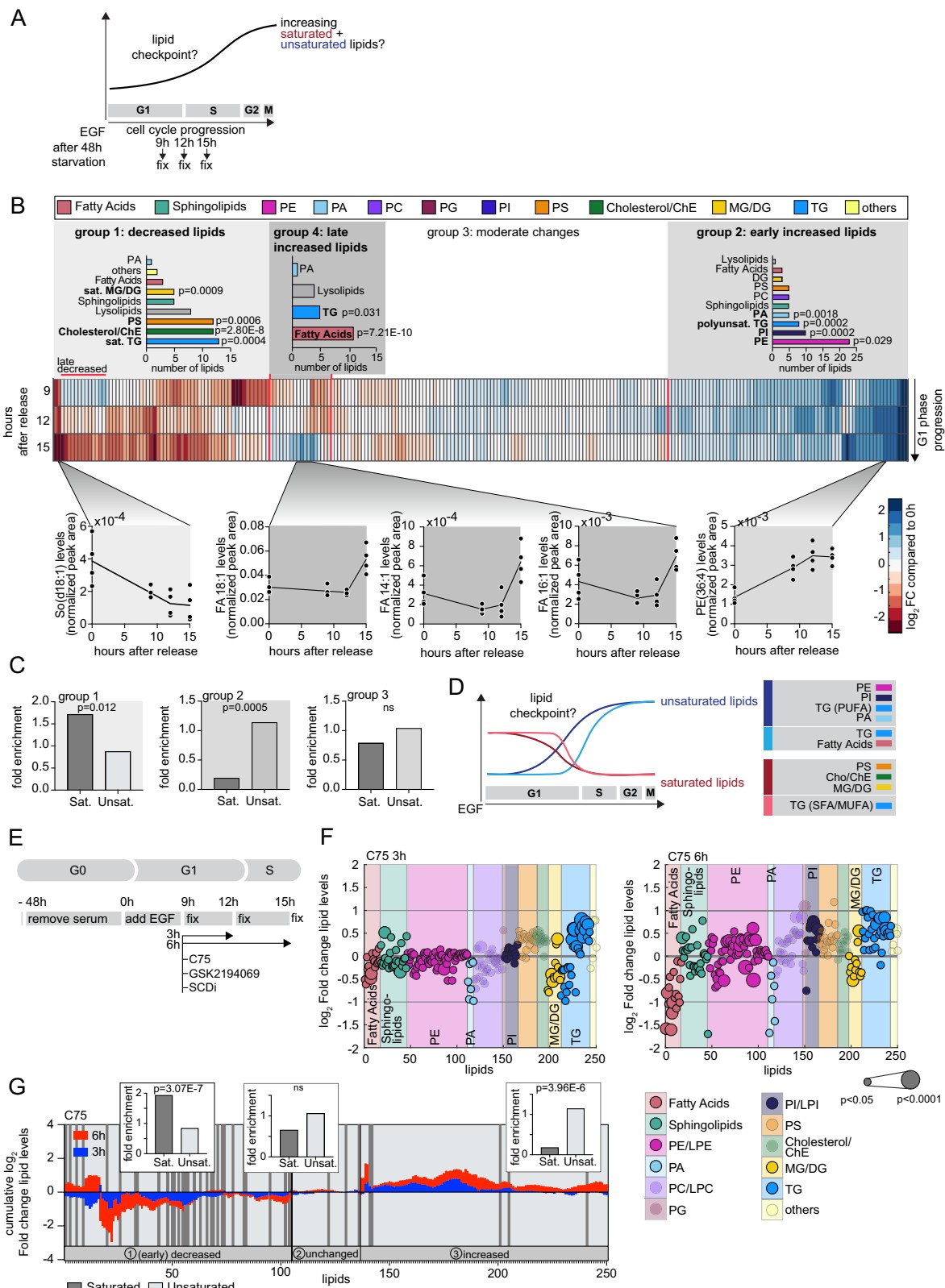

better understand the lipid metabolism network, we performed a lipid-lipid correlation analysis by using lipidome data from all the perturbations we performed at different time points in G1 (Supplementary Data 1). This analysis recapitulated expected positive coregulations, for example the essential ratio[56] between PE and PC species (r = 0.93) (Fig. 3A) or the substrate-product relationship between fatty acids and complex glycerophospholipids (r = −0.95) (Fig. 3B).

To represent the lipid interconnectivity, we transformed the lipid coregulation data calculated between the abundance of the 250 measured lipid species into a network[57] in which the nodes are lipid species and the edges are positive correlation values of r > 0.76 (Fig. 3C, S4A). This network of lipid coregulation at the G1/S transition was circular, displaying the close coregulation of neighboring lipids while simultaneously showing the negative correlations between lipids

**Fig. 2 | The lipidome is reshaped before S phase begins and acute inhibition of FASN induces global lipid changes. A** Diagram depicting cell-cycle phases and workflow. Expected lipid changes and timing of the lipid checkpoint are indicated. **B** Lipidome analysis at different time points of EGF-released MCF-10A cells. Center: Clustergram shows log$_2$ fold-change time course of lipid abundance compared to unreleased cells (blue is increased, red is decreased). Top: 4 groups of lipids based on clustergram (MATLAB). Bar graphs show the number of lipids per class per group. Lipid classes are color-coded. Bottom: Individual lipid examples shown as lipid concentration (normalized peak area) over time. **C** Fold enrichment of saturated (Sat.) or unsaturated (Unsat.) lipids for the individual lipid groups from (**B**). **D** Schematic summary of the data showing early and late increased (blue and light blue) and decreased (red and light red) lipid classes during the cell-cycle progression (summary for S/G2/M is extrapolated based on the data for G1). Lipid classes are color-coded as in (**B**). **E** Schematic depicting workflow of EGF release and inhibitor treatment. **F** Lipidome analysis of cells treated with C75 (30 μM) for 3 or 6 h. Values are shown as log$_2$ fold-change relative to DMSO control treated cells. Each dot represents a lipid species, background and dots are color-coded per lipid class. Dot size indicates significance. Data are combined of four independent experiments and represented as mean. **G** Top: Cumulative log$_2$ fold-change of each lipid species measured after C75 treatment (blue: 3 and red: 6 h). The order of the lipids and the three lipid groups are based on clustergram (MATLAB). Dark gray bars indicate saturated lipid species. Bar graphs show fold enrichment of saturated and unsaturated lipids per lipid group across all 250 lipids measured. **B**, **C**, **G** *P* values are calculated using two-sided Fisher's Exact test for enrichment across 250 lipids. Data are from four independent experiments. FC fold-change, ns not significant, PUFA polyunsaturated, MUFA monounsaturated, SFA saturated.

located on opposite sides of the network (Fig. 3C, S4B). The circularity is likely reflective of the processive metabolic steps that connect precursor and complex lipids. While the precursor lipids including free fatty acids, PA, Cholesterol, and lysolipids (LPC, LPE, LPI) were clustered on one side of the network, the complex lipids including sphingolipids (ceramides, sphingomyelins, sphingosines), glycerophospholipids (PE, PC, PI, PS, PG), and neutral lipids (MG/DG, TG) were located on the opposite side and closely interconnected, reflecting plausible metabolic relationships (Fig. S4C). Markedly, the data-driven coregulation network reconstructed individual metabolic networks such as the TG metabolic pathway. While free fatty acids and PA were tightly coregulated in a local cluster, MG/DG and TG were located further apart on the network likely reflecting the metabolic steps separating these lipids (Fig. 3C).

In conclusion, rather than finding a gradual overall lipid increase, our lipidomic-cell cycle time course and perturbation measurements at the end of G1 identify a striking rapid change of lipids in the TG pathway and re-organization of the overall lipid landscape shortly before the start of S phase which is represented as a circular map of metabolic relationships and coregulation between lipid species.

## Combined lipidome and transcriptome analysis identifies ER stress signaling

Since the lipidome data demonstrated both rapid and global lipid changes close to the G1/S transition and after FASN inhibition, we wondered whether we could link the lipid coregulation network analysis to parallel occurring transcriptional changes and thereby identify the signaling pathway underlying the G1 lipid checkpoint. To do so, we analyzed the transcriptome of the same samples used for lipidome analysis to correlate gene expression changes after acute lipid flux perturbations with lipid abundance changes.

Before analyzing the correlation between the lipidome and large-scale transcriptome data, we validated that this global approach is viable by correlating inhibitor-induced lipid changes to changes in the percentage of S phase entry for each perturbation and time point, using EdU incorporation data. We found that, for example, the levels of polyunsaturated PE (PE(38:5p)) are strongly negatively correlated (r = −0.88) with EdU incorporation while the levels of Palmitoleic Acid (FA 16:1) are positively correlated (r = 0.89) with EdU incorporation (Fig. 4A). Indeed, PE-linked arachidonic acids have been shown to sensitize cells to lipid peroxidation-induced damage[58,59], while monounsaturated free fatty acids such as Palmitoleic Acid are important for cell growth[60].

We then calculated the correlation of each lipid abundance change with the changes in S phase entry for each condition and color-coded the nodes of the lipid network based on these correlation values (Fig. 4B). This mapping revealed that all free fatty acids as well as other metabolically linked precursor lipids were positively correlated with EdU incorporation, predictive of a pro-proliferative role. Interestingly, these precursor lipids were also increased during G1 cell-cycle

progression, including free fatty acids and PA (Fig. 2B, groups 2 and 4). In contrast, other complex lipids that were negatively correlated with EdU incorporation were decreased during G1 cell-cycle progression. These included short-chained saturated TG, MG/DG, and PS species (Fig. 4B (see lipid map on the right) and 2B, group 1). Our correlation analysis comparing lipid changes and S phase entry again points to a plausible role for large-scale ER lipid composition changes regulating the G1 lipid checkpoint.

We next extended the correlation analysis to link the lipid abundance changes to transcriptional changes. We identified all 3739 genes that were differentially regulated after three hours of FASN inhibition using C75 (Supplementary Data 2, 3). Then we calculated the correlation between the transcriptional changes and the lipid abundance changes for each of the 3739 genes across all perturbations. For each differentially regulated transcript we color-coded the lipid nodes on the network according to their correlation value. Such correlations can aid to identify or validate potential roles of metabolic gene expression in regulating the abundance of specific lipids or vice versa. To highlight the general usefulness of the data, *ELOVL7* (Elongation of very long chain fatty acids protein 7) mRNA expression was negatively correlated with short and medium chained free fatty acids and PA levels (Fig. S5A). In order to supply activated long chain fatty acids for lipid synthesis ELOVL7 elongates fatty acids[61]. Other examples of mRNA-lipid regulation include *PLD1* (Phospholipase D1) expression, which is negatively correlated with PC levels, and on the other side of the network, positively correlated with PA levels. PLD1 degrades PC to PA[62]. *HMGCR* (HMG-CoA reductase) expression is negatively correlated with Cho/ChE, consistent with the negative feedback regulation when Cho levels are high[63]. Further, *PTDSS1* (PS Synthase 1) expression is positively correlated with PS levels and negatively correlated with PE/PC, consistent with the reported function[64].

We mapped the correlation of each differentially expressed gene with changes in lipid abundance on the network of 250 lipids and summarized the data in a heat map (Fig. 4C). Among the differentially regulated genes, we identified 11 gene clusters that correlated with different, metabolically linked groups of lipids on the circular network, separating positively and negatively correlated lipids (Fig. 4C and Supplementary Data 2). In this analysis, genes that correlated with similar lipids (directionality on the network) were in the same gene cluster.

We made two critical specific findings about the G1 lipid checkpoint based on this analysis. First, the gene cluster that correlated with the same lipids as the EdU incorporation levels (entry into S phase) contained genes that promote entry into the cell cycle, including the cell-cycle activator *CCND1*, suggesting that lipid changes induced by FASN inhibition in G1 suppress S phase entry by lowering expression of Cyclin D (Fig. 4D). Furthermore, the cell-cycle inhibitor *CDKN1A* (p21) was part of cluster 5 genes which were negatively correlated with lipids that were positively correlated with EdU staining and *CCND1* levels, suggesting that the same lipid changes suppress S phase entry by increasing expression of *CDKN1A* (Fig. 4E).

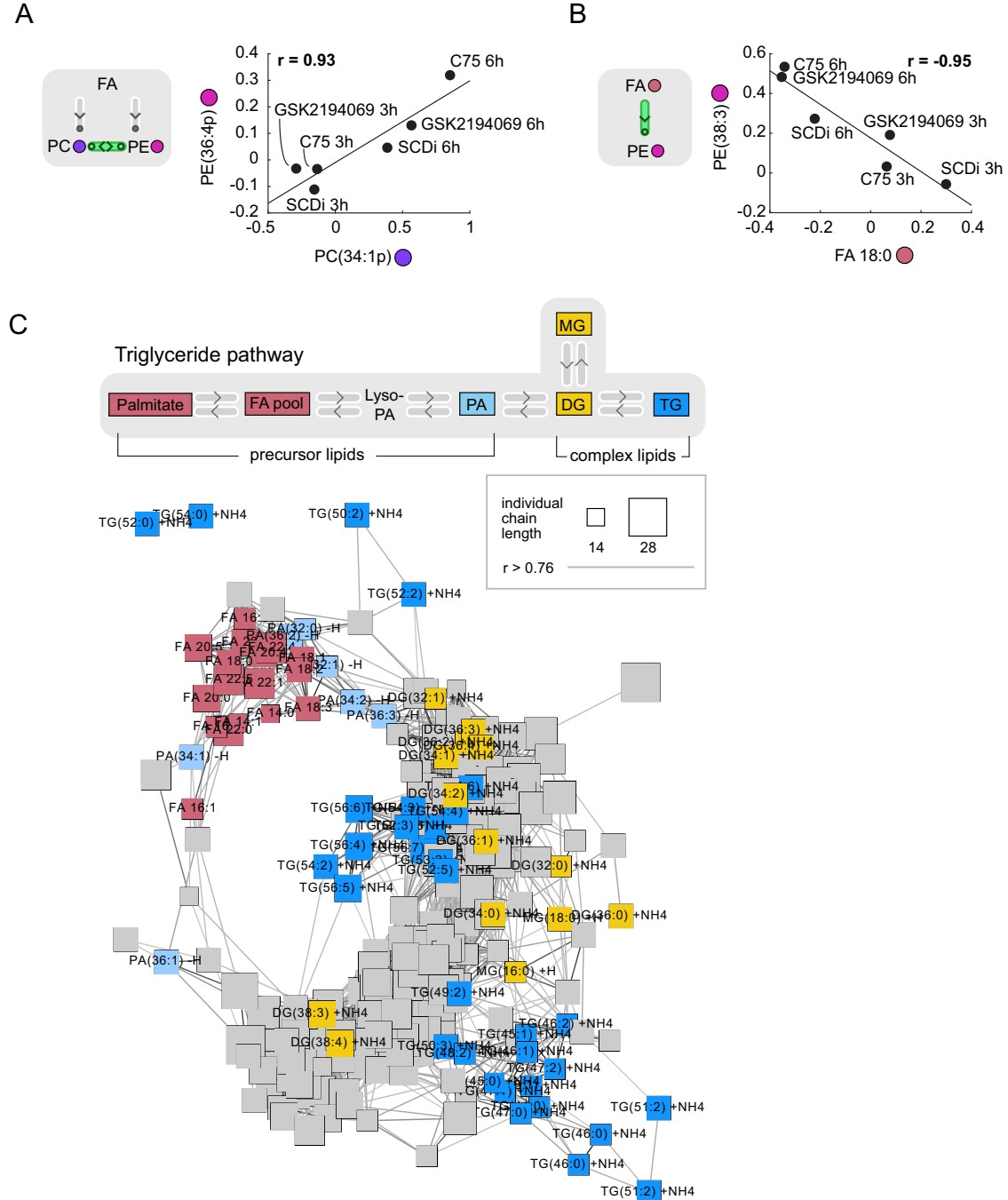

**Fig. 3 | Rapid and global coregulation of precursor lipids and complex lipids in G1.** Scatter plots show example pairs of lipids whose relative abundance over the different inhibitor treatments is positively (**A**) or negatively (**B**) correlated. Lines indicate linear fit. Black dots are labeled based on inhibitor treatment and time point. Schematics show the lipid-lipid relationship based on literature. Colored dots indicate lipid class. Data are combined of four independent experiments and shown as mean. **C** Network visualization of the positive lipid-lipid correlations. Edges are correlations of r >= 0.76. Nodes represent lipid species. Node size represent fatty acid chain length. Lipids in the Triglyceride pathway (see schematic in gray box at top) are color colored and labeled. Gray box highlights lipid relationships based on KEGG pathway. FA fatty acid, FC fold-change, LPA, LPC, LPE, lyso-lipids.

The second main finding was that the cluster 5 genes containing cell-cycle inhibitors were also significantly enriched for genes associated with the PERK (PKR-like eukaryotic initiation factor 2 kinase)-mediated unfolded protein response pathway Gene Ontology (GO) term (38.5-fold enrichment, $p = 0.0025$). The PERK-mediated ER stress response activates ATF4 (activating transcription factor 4) to increase ER stress signaling[65]. Tunicamycin is a bona fide ER stress inducer[66]. Using an expression data set of genes increased upon Tunicamycin stimulation in wild-type mouse embryonic fibroblasts (MEFs) but not

ATF4 knockout MEFs[67], we showed that gene clusters 5 and 6 were significantly enriched for ATF4-dependent genes induced by ER stress (1.7-fold enriched, $p = 0.006$; 1.8-fold enriched, $p = 0.007$, up to 12% of all genes in those clusters) (Fig. 4C, bottom). Indeed, color coding of the lipid-mRNA correlations for *EIF2AK3* (PERK) expression revealed a very similar pattern compared to *CDKN1A* (Fig. 4F). More specifically, in a group of 93 closely clustered genes from cluster 5, we identified the main drivers of the ER stress pathways mediated by IRE1α (inositol-requiring enzyme 1 α)/XBP1 (X-box-binding protein 1) and PERK/ATF4/

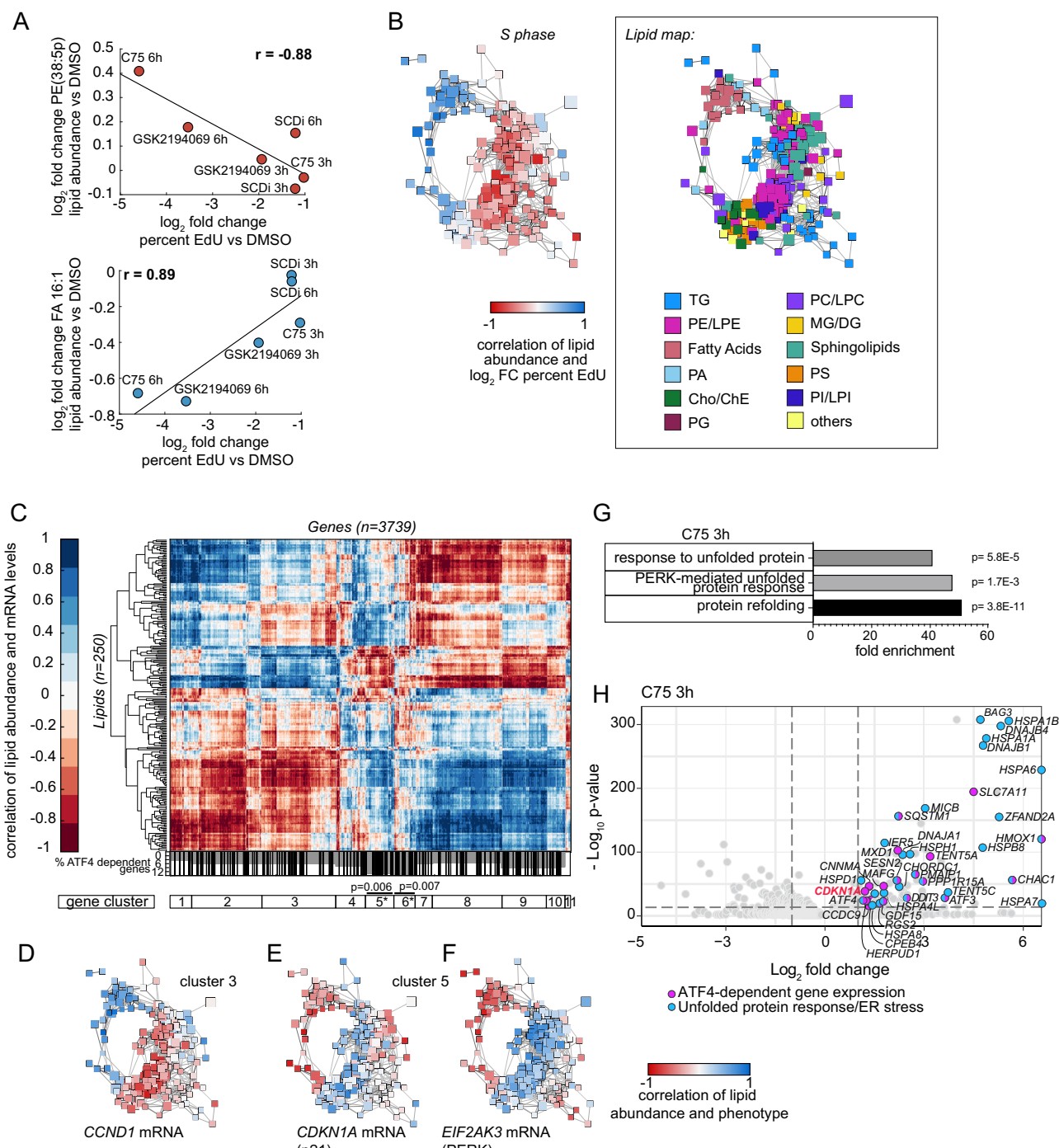

**Fig. 4 | Combined lipidome and transcriptome analysis identifies a link from ER stress and cell-cycle regulators to a G1 checkpoint. A** Scatter plots show example correlations between relative lipid abundance and S phase percentage over the different inhibitor treatments and time points. Line indicates linear fit. Dots are labeled based on inhibitor treatment and time point. **B** Left: Nodes of the network are color-coded based on the correlations between relative lipid abundance and S phase percentage. Right: For orientation the lipid map shows the nodes of the network color-coded by lipid classes. **A**, **B** Data are combined of at least three independent experiments and shown as mean. **C** Top: Hierarchical clustering of lipid-mRNA abundance correlations. Rows: 250 lipids, columns: 3739 differentially expressed genes after C75 treatment. Bottom: Rug plot represents the distribution of ATF4-dependent genes[67] and the gray area shows the percentage of ATF4-dependent genes per gene cluster. 3739 genes are divided into 11 gene clusters (clustergram - MATLAB) based on their lipid correlation. Clusters 5 and 6 are significantly enriched for ATF-dependent genes as calculated by two-sided Fisher's exact test. **D**–**F** Example correlations of selected genes with 250 lipids color-coded on the circular network. Different clusters show different distribution of correlations across the network. **G** Bar plot shows fold enrichment for Gene Ontology (GO) Terms (BP direct) among genes significantly upregulated in cells treated with C75 for 3 h at 9 h after EGF release (mRNA sequencing data, significant in at least 3 replicates, p.adj <0.05, filtered for increased expression in starved cells and at least twofold upregulated compared to DMSO control). *P* values were calculated using EASE score (Fisher's Exact test). Processes sorted by false discovery rate (FDR)–adjusted *P* value. Redundant processes omitted. **H** Volcano plot shows differentially regulated genes after 3 h of C75 treatment. Differentially expressed genes (log$_2$ (fold-change) | > 1, p.adj. <0.014) in blue (genes associated with the ER stress pathway based on GO terms) or magenta (ATF4-dependent genes[67]) or both (blue-magenta). *CDKN1A* (p21) is highlighted in red as the ER stress-cell cycle link. **G**, **H** P. adj. values are Benjamini-Hochberg adjusted p values calculated using the two-sided Wald test. Inhibitor concentrations used: C75 (30 µM), GSK2194069 (50 µM), SCDi (32 µM). FC, fold-change.

CHOP (2.8-fold enrichment, $p = 7.4E{-}05$) (Fig. S5B and Supplementary Data 2). This indicated that the same lipid changes that correlate with increased *CDKN1A* levels and negatively correlate with *CCND1* levels also lead to the induction of an ER stress response. These data demonstrate that mapping of the relationship between lipidome and transcriptome changes (or EdU incorporation rates) onto the lipid coregulation network can identify connections between lipid changes and signaling pathways.

Three different pathways are activated in response to ER stress and are mediated by PERK, IRE1α, and ATF6 (activating transcription factor 6 α). These pathways lead to the rapid activation of transcriptional programs important for coping with ER stress[68–70]. Using the transcriptome data, a GO term analysis identified the ER stress response as a main consequence of acute FASN inhibition since the GO terms "protein refolding", "PERK-mediated unfolded protein response", and "response to unfolded protein" were among the main significantly enriched ones (Fig. 4G, S5C). This finding again suggested a critical role for the rapid induction of a PERK-initiated stress pathway upon lipid perturbation in G1. Many genes that significantly increased after three hours of FASN inhibitor treatment were either associated with GO terms for UPR (unfolded protein response)/ER stress, shown to be ATF4-dependent upon ER stress induction[67], or both (Fig. 4H, S5D). Among all upregulated genes, the ATF4-dependent genes were 2.2-fold significantly enriched ($p = 0.0005$) after treatment with C75 or 2.7-fold significantly enriched ($p = 0.0004$) after treatment with GSKi. The cell-cycle inhibitor *CDKN1A* (p21) was part of this group of genes (Fig. 4H, S5D). These results suggest that the inhibition of FASN and decrease of free fatty acid levels activate a rapid ER stress response that increases the transcription of genes associated with the PERK/ATF4 transcriptional program.

## The rapid G1 arrest is mediated by increased p21 and reduced Cyclin D

Since the acute stress response triggered by FASN inhibition induces an ATF4-dependent transcriptional program, high levels of *CDKN1A* mRNA and low levels of *CCND1* mRNA, we investigated whether there might be a rapid lipid checkpoint signaling pathway mediated by ER stress that increases p21 and reduces Cyclin D1 protein, leading to a cell-cycle arrest in G1. CDK activity in G1 is regulated by several specific, short-lived interactors, most importantly Cyclin D1, p21, and p27[6,71,72]. Cellular stress has been described to increase p21 specifically via the PERK/ATF4 pathway as a direct transcriptional target of ATF4 to increase cell survival during stress[73,74]. Additionally, the PERK-mediated ER stress response induces a G1 cell-cycle arrest by decreasing Cyclin D1 levels[75,76]. To determine how changes in fatty acid availability regulate cell-cycle progression, we measured the nuclear protein levels of the critical regulators of CDK activity after treatment with FASN inhibitors. Indeed, we observed a rapid and dose-dependent increase of the cell-cycle inhibitor p21 and a decrease of the cell-cycle activator Cyclin D1 at both the protein and mRNA level (Fig. 5A–E, S6A, B). Cyclin D1 levels are decreased compared to DMSO treated cells, however they are still increased compared to starved cells (Fig. 5D, E) indicating that the FASN inhibited cells have exited quiescence. As a positive control for p21 induction, we treated cells with Nutlin-3, a p53 activator that induces high p21 levels[77]. Indeed, the increase of p21 levels using Nutlin-3 treatment was comparable to C75 treatment (Fig. 5C, S6C, D).

Significant differences in *CDKN1A* mRNA levels between FASN inhibitor- and control-treated cells were detected as early as two hours after treatment in G1 (Fig. 5E). FASN inhibition did not affect p27 protein levels during this time (Fig. S6E). To confirm that ER stress alters both p21 and Cyclin D1 levels, we tested Tunicamycin treatment which also decreased Cyclin D1 and increased p21 levels, albeit more slowly (after eight hours) than the changes mediated by FASN inhibition (Fig. 5F).

To test whether mammalian Target of Rapamycin (mTOR) inhibition by increased levels of Malonyl-CoA was involved in the lipid checkpoint signaling[78], we measured phosphorylation levels of p70 S6 Kinase (p-T389)[79]. FASN inhibition after 4 or 20 hours of EGF release did not inhibit mTOR signaling as compared to Torin2 treatment (Fig. S6F), a selective mTOR inhibitor[80]. Further, Torin2 treatment during EGF release decreased Cyclin D levels and did not increase p21 levels but rather reduced them after 4 hours, so it did not mimic FASN inhibition (Fig. 5C, S6G). To test if mitochondrial function was important for the lipid checkpoint, we inhibited fatty acid oxidation with Etomoxir or oxidative phosphorylation with Oligomycin A[81]. Both treatments did not increase p21 levels after 4 hours during EGF release (Fig. S6H, I). Etomoxir treatment did not decrease Cyclin D levels while Oligomycin A did (Fig. S6H, I).

## FASN inhibition causes loss of Rb phosphorylation in G1

Before cells enter S phase and DNA replication is initiated, CDK4/6 activity is often required to phosphorylate Rb to activate the E2F transcriptional response (Fig. 6A)[46,51,52]. Treatment with FASN inhibitors or siRNAs targeting FASN resulted in a dose-dependent decrease of Rb phosphorylation at S807/S811 (p-S807/S811) (Fig. 6B, C, S6J–M), explaining how lipid stress increases Cyclin D and decreases p21 to block CDK4/6 activity, which prevents Rb phosphorylation and inhibits E2F-mediated transcription. The absence of Rb phosphorylation and DNA replication was reversed by supplementing cells with serum lipids (Figs. 6D, 1E). Additionally, supplementation of cells with Palmitate complexed to BSA rescued Rb (p-S807/S811) levels in the presence of FASN inhibition even though fatty-acid free BSA and Palmitate-BSA alone reduced the levels of Rb (p-S807/811) positive cells (Fig. 6E). These data demonstrate that inhibition of FASN blocks S phase entry by disrupting Palmitate and lipid synthesis, and deregulating Rb signaling. Again, treatment with Tunicamycin confirmed that ER stress suppresses Rb (p-S807/S811) phosphorylation and EdU incorporation (Fig. 6F, S6N).

To further confirm that other pathways leading to reduced free fatty acids also triggered this G1 checkpoint response, we used BEL to inhibit phospholipase activity. Indeed, BEL treatment for four hours after EGF release also induced p21 and reduced Cyclin D levels (Fig. S7A). We also observed a dose-dependent reduction of Rb (p-S807/S811) after 20 hours EGF release and BEL treatment (Fig. S7B). To show that the metabolic pathway upstream of FASN also impinges on p21 and Cyclin D, EGF-released cells were treated with the ACACA inhibitor TOFA. Indeed, p21 levels were increased and Cyclin D levels were decreased after 4 hours (Fig. S7C).

Inhibition of SCD which acts downstream of FASN-synthesized fatty acids, however, did not change p21 or Cyclin D levels and only moderately reduced Rb (p-S807/811) levels after 20 hours of EGF release (Fig. S7D, E), suggesting that reducing unsaturated fatty acids does not trigger this specific cell cycle signaling pathway leading to the G1 checkpoint response.

To confirm that ER stress contributed to the decrease of Rb (p-S807/811) phosphorylation when FASN was inhibited, we used an inhibitor targeting PERK activation (Perki, GSK2606414)[82]. Indeed, there was an increase in Rb (p-S807/811) positive cells when PERK was inhibited compared to cells treated with FASN inhibitor alone (Fig. S7F).

Finally, to confirm that the increased levels of p21 contribute to the decrease in the level of Rb (p-S807/S811) phosphorylation in FASN inhibitor-treated cells, we used siRNA targeting *CDKN1A*. Supporting the finding that the lipid checkpoint acts by upregulating p21, there was a reduced response to FASN inhibitor treatment in cells deficient for p21 (Fig. 6G). Decreased levels of p21 led to an increase of EdU incorporation levels compared to control cells treated with increasing doses of FASN inhibitor (Fig. S7G, H). As p53 is an important regulator of p21 levels and G1 checkpoint regulation[83], we next measured p53

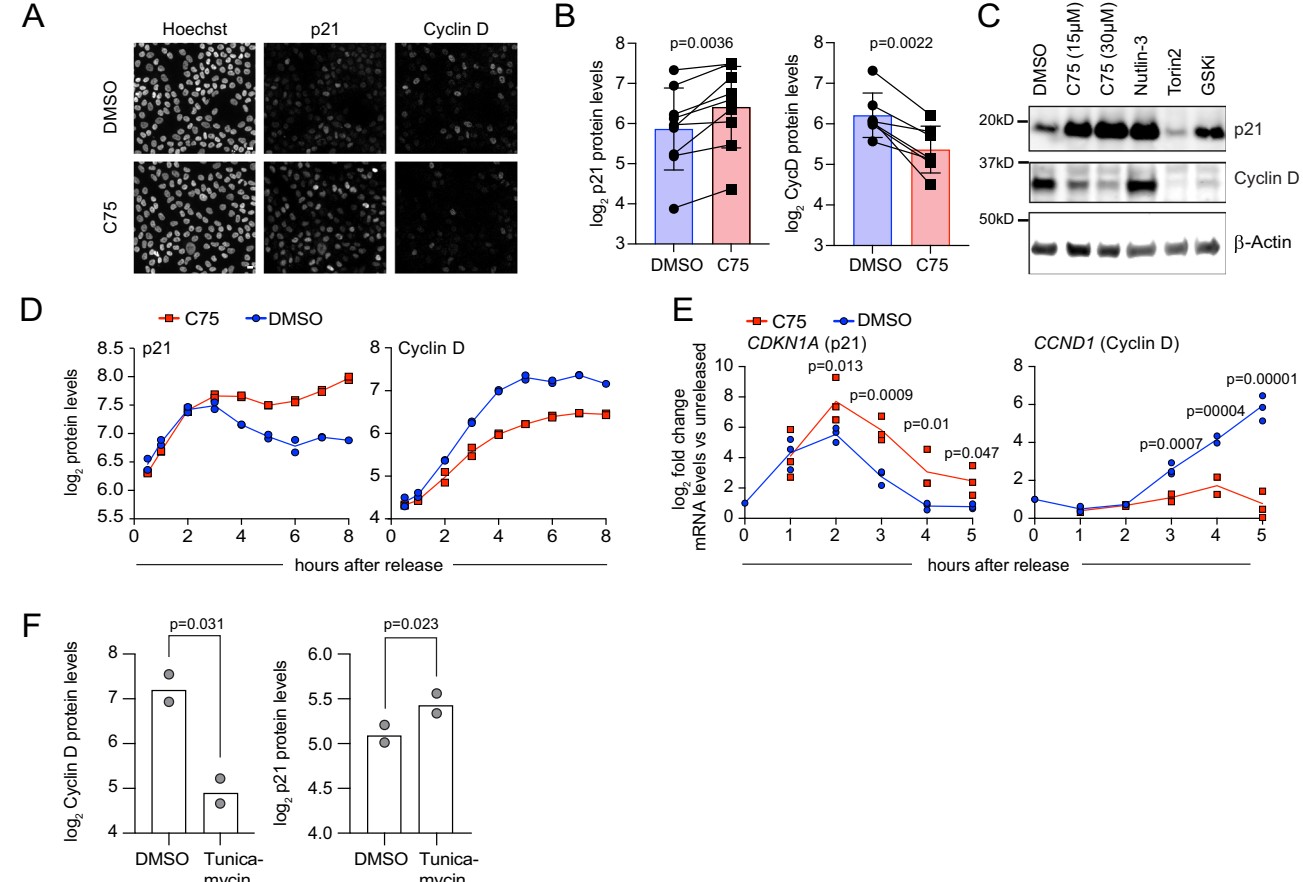

**Fig. 5 | Inhibition of FASN triggers a rapid G1 arrest mediated by increased p21 and reduced Cyclin D. A** EGF-released MCF-10A cells (4 h) treated with DMSO or C75. Sample immunofluorescence images for nuclei (Hoechst), p21, and Cyclin D are shown. Scale bar: 10 μm. Data are representative of at least seven independent experiments. **B** Quantification of nuclear protein levels as shown in (**A**). Data are shown as mean ± SD from at least seven independent experiments, n > 20,000 cells per condition. **C** EGF-released cells (4 h) treated with indicated inhibitors. Blotted for Cyclin D, p21, and β-actin (loading control). Data are representative of three independent experiments. **D** Nuclear p21 and Cyclin D protein levels after EGF release and treatment with DMSO or C75 measured by immunofluorescence. Data are representative of at least two independent experiments (data points are means

of two technical replicates), n > 17,000 cells per condition. **E** mRNA levels of *CDKN1A* (p21) and *CCND1* (Cyclin D) measured by qRT-PCR after treatment with DMSO or C75 and normalized to unreleased cells. Data are from at least two independent experiments. **F** Cyclin D and p21 nuclear protein levels measured by immunofluorescence after 8 h of EGF release in the presence of DMSO or Tunicamycin. Data are from two independent experiments, n > 12,000 cells per condition. **B**, **E**, **F** *P* values calculated using two-tailed Paired *t* test. Inhibitor concentrations used unless indicated otherwise: C75 (15 μM), GSK2194069 (50 μM), Tunicamycin (10 μg/ml), Nutlin-3 (10 μM), Torin2 (500 nM). Source data are provided as Source data file.

levels in FASN inhibitor-treated cells. Protein levels were unchanged and mRNA levels were slightly reduced (Fig. S7I, J). Additionally, MCF-10A cells deficient for p53 treated with FASN inhibitor still showed reduced EdU incorporation, Rb (p-S807S/S811), and Cyclin D levels, and increased p21 levels similar to WT cells (Fig. S7K). These results argue that altered G1 signaling rapidly induced by FASN inhibition is p53 independent.

## The G1 arrest mediated by the lipid checkpoint is reversible

Our data so far provide a mechanistic basis how cells trigger an acute and global G1 cell-cycle arrest that fulfills the first two criteria of a lipid cell-cycle checkpoint. We next directly tested the third required checkpoint criterion: Whether the checkpoint-induced cell-cycle arrest is reversible after FASN inhibitor removal. To test this, we released MCF-10A cells with EGF in the presence of FASN inhibitors or Tunicamycin. The media was replaced after 20 hours and EdU incorporation was measured after another 24 hours (Fig. 6H).

Markedly, the inhibition of S phase entry by FASN inhibitors was fully reversible after the inhibitor removal and the percentage of EdU positive cells was even slightly higher compared to the control cells. In contrast, Tunicamycin-treated cells only showed a limited 5.5%

recovery from the drug treatment after removal (Fig. 6H). We speculate that this difference is because the FASN inhibitor-treated cells change ER lipids which then trigger a rapid ER stress response. This lipid stress response stops as soon as the normal lipid composition is restored. The slow kinetics in response to Tunicamycin are likely explained by a slow accumulation of unprocessed proteins upon blockage of the glycolytic protein linkages. This possibly leads to a slower recovery of the unprocessed proteins after the blockage is removed. Thus, the lipid-induced ER stress response has a unique kinetic feature in that it can rapidly start and stop, which is consistent with a G1 checkpoint response but is distinct from the more commonly studied ER stress response that is associated with protein misfolding and cannot readily be reversed.

We conclude that inhibition of fatty acid synthesis in G1 leads to a massive and acute change in the ER lipid composition. This change in lipid composition induces the PERK/ATF4 ER stress pathway, increases p21 levels, decreases Cyclin D1 levels and phosphorylation of Rb (p-S807/S811), and thereby prevents cells from entering S phase without causing cell death (Fig. 6I). Markedly, lipid stress is rapidly reversed, and cells enter S phase after fatty acids become available again.

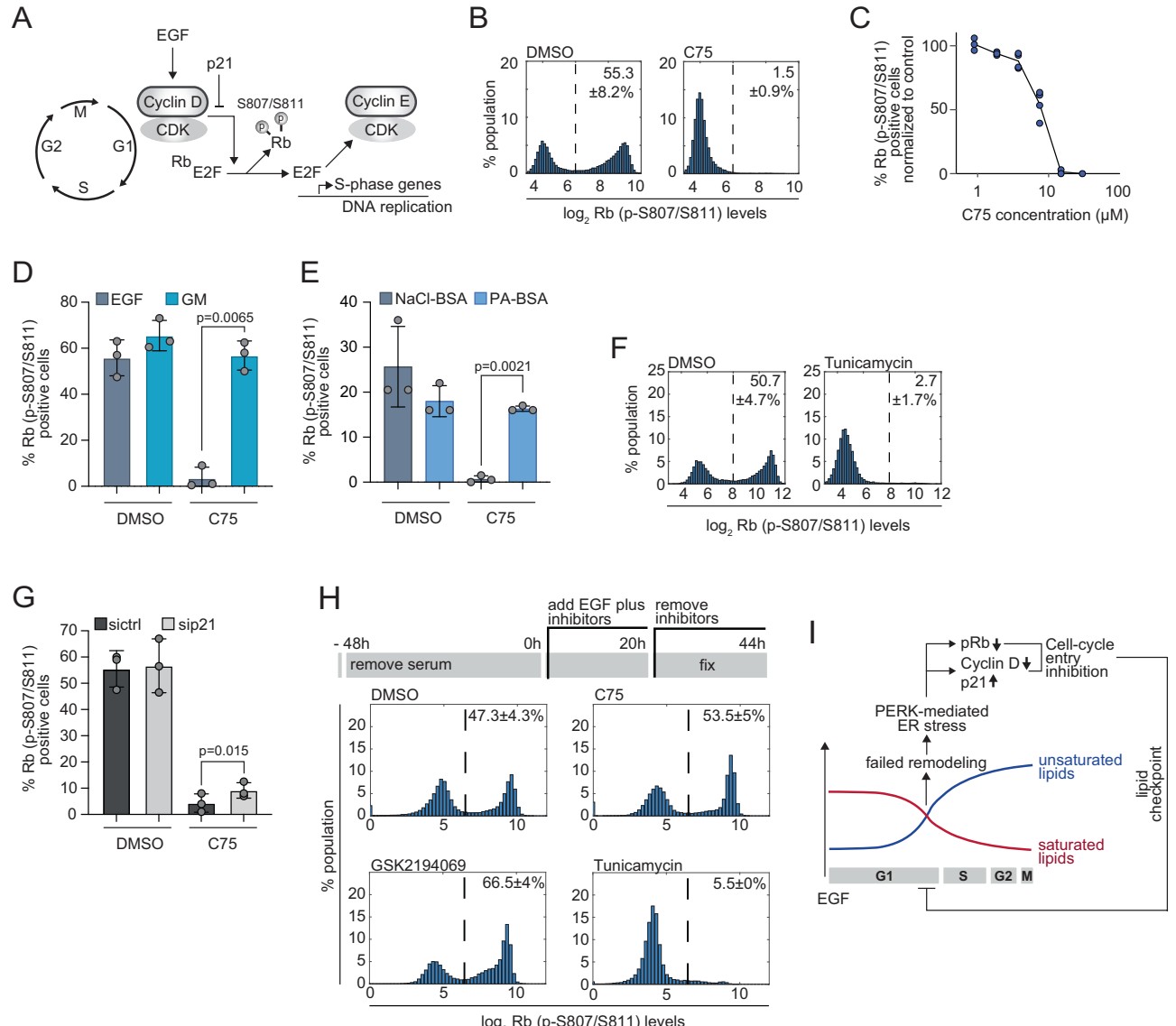

**Fig. 6 | The G1 arrest mediated by the lipid checkpoint leads to loss of Rb phosphorylation and is reversible. A** Schematic highlighting Cyclin-dependent kinase (CDK) signaling in G1. **B** Histogram of Rb(p-S807/S811) signal of cells treated with C75 or DMSO after EGF release (20 h). Percentages are shown as mean ± SD of four independent experiments, n > 17,000 cells per condition. **C** Dose response of percent Rb(p-S807/S811) positive cells treated with C75 normalized to DMSO treatment after EGF release (20 h). Data are from at least three independent experiments, n > 17,000 cells per condition. **D** Percentage of Rb(p-S807/S811) positive cells treated with DMSO or C75 after EGF release (20 h). GM (growth media): 5% serum. Data are shown as mean ± SD from three independent experiments, n > 15,000 cells per condition. **E** Percentage of Rb(p-S807/S811) positive cells treated with DMSO or C75 after EGF release (20 h). PA-BSA indicates the presence of Palmitate complexed to BSA. NaCl-BSA is the control treatment. Data are shown as mean ± SD from three independent experiments, n > 15,000 cells per condition. **F** Histogram of Rb(p-S807/S811) signal of cells treated with Tunicamycin or DMSO and EGF-released (20 h). Percentages are shown as mean ± SD of at least

three independent experiments, n > 15,000 cells per condition. **G** Percent of Rb(p-S807/S811) positive cells transfected with sicontrol (sictrl) or sip21 and treated with DMSO or C75. Data are shown as mean ± SD from three independent experiments, n > 14,000 cells per condition. **H** Schematic depicting the workflow of treatments. Histogram of Rb(p-S807/S811) signal of cells treated with different inhibitors. Percentages are from two independent experiments and shown as mean ± SD, n > 20,000 cells per condition. **I** Schematic summary of the findings: Lipidome analysis shows the balance of increased unsaturated lipids (blue) and decreased saturated lipids (red) to overcome the lipid checkpoint in G1 (outlook for S/G2/M is extrapolated based on the G1 data). Lipid checkpoint engagement mediated by PERK-mediated ER stress feeds back to cell-cycle signaling by increasing p21 levels, decreasing Cyclin D and Rb (p-S807/S811) causing a cell-cycle delay. **D, E, G** P values calculated using two-tailed Paired t test. Inhibitor concentrations used unless indicated otherwise: C75 (15 μM), GSK2194069 (50 μM), Tunicamycin (10 μg/ml), Palmitate (5 μM). Source data are provided as Source data file.

## Discussion

Our lipidome analysis shows that cells enter the cell cycle by a massive change in their lipid composition in G1, and that fatty acid synthesis is rate-limiting for cells to start S phase when external fatty acid availability is low. Our single-cell imaging analysis shows that insufficient fatty acids suppress the lipid remodeling program and trigger a lipid checkpoint in G1, mediated by ER stress-initiated PERK/ATF4 signaling

and Cyclin D/p21/Rb-mediated suppression of E2F activation and S phase entry. The consequences of inhibiting FASN activity fulfill the three requirements of a cell-cycle checkpoint. First, lowering fatty acid synthesis stops entry into the cell cycle by inducing an ER stress signaling program linked to the suppression of Rb phosphorylation and E2F activation. Second, the acute transcriptional response triggered by loss of fatty acids is fast-acting as changes in the level of CDK

regulators are detected as early as two hours after FASN inhibition. Third, removal of the FASN inhibitor, addition of serum lipids or Palmitate rapidly reverses the cell-cycle arrest and cells enter S phase. Markedly, the lipid checkpoint is triggered already in G1, long before cells critically require lipid biosynthesis for a successful mitosis. Indeed, live cell analysis confirms that reduced free fatty acid levels in G2 and mitosis can cause mitotic defects. Fatty acids that are synthesized late in the cell cycle are important for nuclear envelope formation[84], possibly explaining why FASN inhibition in G2 can lead to mitotic defects. In this way, cells sense if there is insufficient lipid biosynthesis and halt the cell cycle by employing an anticipatory G1 lipid checkpoint that prevents the start of a typically over 12 hour-long process that would end in a defective mitosis when lipid composition is most critical. This lipid checkpoint is likely generally used by non-transformed cells while many cancer cells have lost the G1 lipid checkpoint because of mutated Rb, p21, or upregulated CDK4/6 activity.

We started our study expecting that most lipid species simply increase in G1 but instead found a remodeling of complex lipids at the beginning of the cell cycle. We also expected that FASN inhibition simply stops lipid synthesis but found not only an overall rapid reduction of certain lipid classes such as fatty acids and PA but also lipid remodeling resulting in the relative increase of lipid species including TG. A plausible interpretation of our results is that mitogens must trigger an increase in lipid abundance and remodeling in G1 to overcome a lipid checkpoint response before cells can enter S phase. Our study argues that cells sense lipid synthesis during the cell cycle by using the same ER stress pathway that senses protein misfolding and protein crowding but by monitoring a broad panel of complex ER regulatory lipids at the G1/S transition, including lipids in the PE, PA, MG/DG, and TG classes. This interpretation is supported by earlier studies that investigated long term changes in lipid composition after genetic or chemical perturbations that resulted in ER stress responses[85–90] and a recent study showing that cell-cycle dependent ER membrane biogenesis protects against mitotic chromosome missegregation[91]. Our data further showed that FASN inhibition leads to a rapid increase in long-chained unsaturated lipids including TG species which supports a proposed protective role of the PERK/ATF4 pathway in promoting TG lipogenesis[92–95]. A shift from saturated to unsaturated lipid species increases membrane fluidity[96,97], which may be important for a successful completion of mitosis and to promote the increased size of the ER during cell-cycle progression. Other lipid changes such as long-term treatment with Palmitate induce an "excess lipid" ER stress response[98,99].

Finally, our study shows how to use lipidomic data as a discovery tool to learn how lipid changes regulate signaling processes. Since lipids are metabolically and structurally linked, identifying how individual lipids control cell function is challenging. By combining different perturbations and time points with transcriptome and lipidome coregulation analysis, we introduce a framework using the unbiased annotation of lipid changes and function to identify associated signaling processes. Our identification and characterization of a G1 lipid checkpoint demonstrates how the combined lipidomic and transcriptomic approach can uncover links from lipid changes to signaling pathways.

## Methods
### Cell culture
All experiments were performed using MCF-10A cells at 30–80% confluency. MCF-10A cells (ATCC, CRL-10317) were cultured in phenol red-free DMEM/F12 (Invitrogen) supplemented with 5% horse serum (ATCC), 20 ng/mL EGF (PeproTech), 10 μg/mL insulin (Sigma-Aldrich), 0.5 μg/mL hydrocortisone (Sigma-Aldrich), 100 ng/mL cholera toxin (Sigma-Aldrich), 50 U/mL penicillin and 50 μg/mL streptomycin (Thermo Fisher). MCF-10A p53−/− cells were obtained from Horizon

Discovery (HD 101-005). For starvation, cells were cultured for 2 days in starvation media (phenol red-free DMEM/F12 (Invitrogen)) supplemented with 0.5 μg/mL hydrocortisone (Sigma-Aldrich), 100 ng/mL cholera toxin (Sigma-Aldrich), 0.3% bovine serum albumin (Sigma-Aldrich), 50 U/mL penicillin and 50 μg/mL streptomycin (Thermo Fisher). To release into the cell cycle with EGF, cells were cultured in starvation media and 20 ng/ml EGF (PeproTech). RPE1-hTERT human retinal pigment epithelial cells (ATCC, CRL-4000) were cultured in DMEM/F12 plus 10% FBS and 0.01 mg/mL hygromycin B. RPE1 were starved in DMEM/F12 and released with DMEM/F12 plus 0.5% FBS.

### Constructs and stable cell lines
MCF-10A cells were transduced with lentiviral vectors encoding CSII-pEF1a-H2B-mTurquoise, CSII-pEF1a-DHB (aa994-1087)-mVenus, and CSII-pEF1a-mCherry-Geminin(aa1-110)[3,47]; lentiCas9-Blast was a gift from Feng Zhang (Addgene plasmid #52962).

### RNA-Seq and analysis
RNA was extracted using Qiashredder and RNeasy Kit (Qiagen) according to manufacturer's instructions and quality control was performed using an Agilent Bioanalyzer 2100. Sequencing library preparation was performed at the Stanford Functional Genomics Facility using the KAPA Stranded mRNA-Seq Kit and mRNA Capture Beads (Roche) according to manufacturer's instructions. Single-index 8-mer adapters were added, and the libraries were sequenced with 75 bp single-end reads on an Illumina NextSeq platform.

Raw reads were mapped onto a human genome assembly (GRCh38) using TopHat2[100] and assigned to individual genes using featureCounts[101]. Differential expression analysis was performed using DESeq2 (R Studio 1.1.463)[102] (Supplementary Data 3).

### qRT-PCR
RNA was reverse transcribed using oligo dT primers and RevertAid Reverse Transcriptase (both Thermo Fisher). qRT-PCR was performed using iTAQ Universal SYBR Green Supermix (BioRad) in technical triplicates analyzed on a Roche Lightcycler480 II. Results were normalized to the housekeeping gene EEF1A1. Primer sequences: EEF1A1_fw: GATGGCCAGTAGTGGTGGAC, EEF1A1_rev: TTTTTCGCAACGGGTTTG, CDKN1A_fw: CATGGGTTCTGACGGACAT, CDKN1A_rev: AGTCAGTTCC TTGTGGAGCC, CCND1_fw: GGCGGATTGGAAATGAACTT, CCND1_rev: TCCTCTCCAAAATGCCAGAG.

### Lipidomics
Lipids were extracted using a modified version of the Bligh-Dyer method[103]. Briefly, samples were manually shaken in a glass vial (VWR) with 1 mL PBS, 1 mL methanol and 2 mL chloroform containing internal standards ($^{13}C16$-palmitic acid, d7-Cholesterol) for 30 s. The resulting mixture was vortexed for 15 s and centrifuged at 2400 x g for 6 min to induce phase separation. The organic (bottom) layer was retrieved using a Pasteur pipette, dried under a gentle stream of nitrogen, and reconstituted in 2:1 chloroform: methanol for LC/MS analysis.

Lipidomic analysis was performed on a Vanquish HPLC online with a Q-Exactive quadrupole-orbitrap mass spectrometer equipped with an electrospray ion source (Thermo Fisher). Data was acquired in positive and negative ionization modes. Solvent A consisted of 95:5 water: methanol, Solvent B was 60:35:5 isopropanol: methanol: water. For positive mode, solvents A and B contained 5 mM ammonium formate with 0.1% formic acid; for negative mode, solvents contained 0.028% ammonium hydroxide. A Bio-Bond (Dikma) C4 column (5 μm, 4.6 mm × 50 mm) was used. The gradient was held at 0% B between 0 and 5 min, raised to 20% B at 5.1 min, increased linearly from 20% to 100% B between 5.1 and 55 min, held at 100% B between 55 min and 63 min, returned to 0% B at 63.1 min, and held at 0% B until 70 min. Flow rate was 0.1 mL/min from 0 to 5 min, 0.4 mL/min between 5.1 min and 55 min, and 0.5 mL/min between 55 min and 70 min. Spray voltage

was 3.5 kV and 2.5 kV for positive and negative ionization modes, respectively; S-lens RF level is 50. Sheath, auxiliary, and sweep gases were 53, 14 and 3, respectively. Capillary temperature was 275 °C. Data was collected in full MS/dd-MS2 (top 5). Full MS was acquired from 150–1500 m/z with resolution of 70,000, AGC target of $1 \times 10^6$ and a maximum injection time of 100 ms. MS2 was acquired with resolution of 17,500, a fixed first mass of 50 m/z, AGC target of $1 \times 10^5$ and a maximum injection time of 200 ms. Stepped normalized collision energies were 20, 30 and 40%.

Lipid identification was performed with LipidSearch (Thermo Fisher). Lipids were annotated at the species level. The list of LipidSearch-identified lipids was transferred to Skyline[104], where mass accuracy, chromatographic retention time and peak integration of all lipids were manually verified. Peak intensities normalized by total signal were used for lipid quantitation.

## Immunoblotting
Cells were plated in a 60 mm diameter dish, starved, and released with EGF for 4 h or 20 h in the presence of indicated drugs. At those time points, cells were washed with cold PBS then 80 μl of RIPA buffer (Cell Signaling Technology) supplemented with Protease and Phosphatase Inhibitor Cocktail was added before the cells were scraped off the plate. Membrane and DNA were sheared in the samples released for 20 h using a 25 G syringe. All samples were spun down at 4 °C for 15 min at 13,000 rpm. Protein concentrations were determined using the Pierce BCA Protein Assay kit (Thermo Scientific). Samples were diluted to 14 μg/μl protein concentration and Laemmli SDS sample buffer (Boston BioProducts, Inc.) was added. Samples were separated by SDS-PAGE using Bolt 4–12% Bis-Tris Plus protein gels and Bolt MES SDS running buffer (both Invitrogen) for the 4-hour samples (smaller proteins) or NuPAGE MOPS SDS running buffer (50 mM MOPS, 50 mM Tris Base, 0.1% SDS, 1 mM EDTA, pH 7.7) for the 20-hour samples. Proteins were transferred onto PVDF membranes (Immobilon, EMD Millipore) using wet transfer in transfer buffer + 10% methanol. The membranes were blocked with blocking buffer (LI-COR) for 1 h at room temperature. Antibodies were added in TBST + 5% BSA + 0.01% NaN3 at 4 °C overnight. After washing the membrane with TBST, secondary antibodies conjugated to Alexa Fluor −680 and −800 (LI-COR) were added for 1 h at room temperature. After washing with TBST, the membranes were scanned using an Odyssey Infrared Imaging System (LI-COR Biosciences). Uncropped scans of the blots are present in the Source data file and in the Supplementary Information file.

## Immunofluorescence
Cells were fixed in 4% paraformaldehyde, permeabilized in 0.2% Triton X-100 (in PBS) for 15 min and blocked with blocking buffer (10% fetal bovine serum, 1% bovine serum albumin, 0.1% Triton X-100, and 0.01% NaN₃ in PBS) for 1 h at room temperature. Primary antibodies (diluted in blocking buffer) were added overnight at 4 °C. Cells were washed three times with PBS and stained with secondary antibodies conjugated to Alexa Flour −488, −568, or −647 (Thermo Scientific) for 1 h. Cell nuclei were stained with Hoechst 33342 (Thermo Fisher Scientific) for 10 min and washed three times with PBS. For EdU incorporation, cells were treated with 10 μM EdU (Thermo #A10044) for 15 min at 37 °C. To label EdU, cells were treated with Click reaction solution (2 mM CuSO4, 25 mg/mL sodium ascorbate, 3 μM AF647-picolyl-azide (Click Chemistry Tools 1300) in 1x TBS, pH 8.3) for 20 min at room temperature as previously described[105]. Unless otherwise indicated MCF-10A cells were EGF released for 20 h in the presence of control or inhibitor treatments prior to EdU pulse and fixation.

## siRNA and sgRNA
siRNAs or sgRNAs and tracrRNA (trans activating CRISPR RNA, U-002005-20) (all from Dharmacon) were transfected into cells using DharmaFect 1 (Dharmacon) according to manufacturer's instruction at a final concentration of 20 nM. SgRNAs and tracrRNAs were transfected into MCF-10A cells stably expressing Cas9 (Supplementary Table 1). Two non-targeting control crRNAs were used (U-007502-01 and U-007503-01). Cells were transfected and starved simultaneously. After 48 h, cells were released with EGF for 20 h, pulsed with EdU for 15 min and fixed, stained, and imaged. For knockout efficiency, we stained with FASN antibody and quantified all cells below the cutoff (log₂ FASN protein levels =7) (Fig. S1I). Pools of four individual siRNAs were used to target p21 and FASN and compared to pools of non-targeting siRNAs (Supplementary Table 2).

## Palmitate addition
MCF-10A cells were starved for 48 h. C75 and DMSO were added in the presence of EGF and Palmitate (Sigma Aldrich) complexed to fatty-acid free BSA (Sigma Aldrich). Palmitate-BSA was prepared by dissolving Sodium Palmitate in 150 mM NaCl solution at 70 °C making a 1 mM Palmitate solution. This was added to a BSA-NaCl solution at a 6:1 ratio at 40 °C. As a control we used BSA-NaCl solution without Palmitate. The final concentration of Palmitate-BSA used equaled 5 μM Palmitate and the corresponding NaCl-BSA concentration was used as a control.

## Imaging
MCF-10A cells were either plated on plastic (Corning 3904) or glass-bottom 96 well plates (Cellvis P96-1.5H-N), coated with 20 μg/mL bovine plasma fibronectin (Sigma-Aldrich F1141). Images were acquired on an ImageXpress Micro and Micro Confocal microscope (Molecular Devices) or an ECLIPSE Ti2 inverted microscope (Nikon) in a 37 °C humidified chamber with 5% CO₂. A 10x objective (0.3 N.A.) with no binning was used for most fixed-cell immunofluorescence and for live-cell imaging of H2B, cyclin E/A-CDK activity reporter, and APC/C^Cdh1 degron reporter. For fixed-cell puncta immunofluorescence, a 20x objective (0.75 N.A.) with 2-by-2-pixel binning was used. The MetaXpress software (Molecular Devices) and an Andor Zyla sCMOS camera were used for image acquisition. During live-cell imaging, cells were imaged in growth media and images were taken every 12 min or 15 min, with the total light exposure being under 300 ms for each multi-color image. Fixed cells were imaged in PBS.

## Image analysis
Image analysis was performed with a custom MATLAB (R2020a) pipeline[3,46]. Briefly, optical illumination bias was corrected for by acquiring the background autofluorescence signal across all wells, which was used to flatten all the images. Then, a global background was subtracted from every image. Nuclei were segmented using the H2B-mTurquoise (for tracking during live-cell imaging) or Hoechst (fixed-cell imaging) signal. To measure the cyclin E/A-CDK activity reporter signal, a cytoplasmic area in the shape of a ring around the nucleus was sampled (with inner radius of 0.65 μm and outer radius of 3.25 μm) without overlapping with cytoplasm from a neighboring cell. The activity was then calculated by taking the ratio between the median cytoplasmic intensity and the median nuclear intensity. For quantification of fixed-cell images, DNA content was calculated as the total Hoechst intensity. Nuclear fluorescence signals were calculated as the median intensity. Nuclear puncta (γH2A.X p-S139) were identified as the foreground by thresholding on background-subtracted images and puncta count was the number of foreground pixels. For cytoplasmic quantification the shape of a ring around the nucleus was quantified (with inner radius of 0.65 μm and outer radius of 3.25 μm) without overlapping with cytoplasm from a neighboring cell. ImageJ2 (Fiji) 2.3.0/1.53 f was used for example images and scale bars.

## Lipidomic analysis
**Normalization.** The lipidomics results were normalized based on the sum of concentrations for all lipid species measured in a single

biological replicate. Next, values were averaged over the four biological replicates, and log$_2$ transformed against the corresponding average concentrations measured in the DMSO-treated and time-matched sample.

**Coregulation network analysis.** Lipid coregulation was defined as correlation values of 0.76 or higher. Various natural network layout algorithms (in Cytoscape and R) were tested to confirm that the circularity shown was not based on a network layout optimization algorithm. The threshold of 0.76 was tested to be sufficiently robust to leaving out the single treatment that contributed most to any one correlation value.

**Lipid-gene correlation and clustering.** All 3739 differentially expressed genes after 3 h of C75 treatment compared to the according DMSO treatment (cutoff: log$_2$ fold change >1, FDR < 0.1) were used to calculate the lipid-gene correlation. The changes of these transcripts were measured in the presence of different lipid metabolism inhibitors (Supplementary Data 2). Lipid values were the log$_2$ fold changes compared to the DMSO-treated and time-matched sample (Supplementary Data 1). These data sets were then used to calculate the correlation values (Supplementary Data 2) and clustered based on their similarities using the clustergram function in Matlab.

### Statistics
If not stated otherwise, statistical analyses were performed using Student's two-sample t test or paired/one-sample t test, or Fisher's exact test (MATLAB fishertest). Quantifications are represented as mean ± standard deviation. Further details and p values can be found in the Fig. legends. Volcano plots were generated using EnhancedVolcano (https://github.com/kevinblighe/EnhancedVolcano) in R.

### Antibodies
Rb (p-807/811) (#8516, (IF: 1:2500, Western blot (WB): 1:1000)), Rb (#9309, IF: 1:1000), p21 (#2947, WB: 1:1000), p27 (#3686, IF: 1:1600), p53 (#2527, IF: 1:1600), γH2A.X (p-S139) (#2577, IF: 1:500), p70 S6Kinase (p-T389) (#9234, WB: 1:200), β-Actin (#8457, WB: 1:2000), (all Cell Signaling Technology); Rb (#554136, WB: 1:250), p21 (Clone SXM30, mouse, # 556431, IF: 1:250, both BD Biosciences); Cyclin D1(SP4) (#MA5-16356, IF and WB: 1:500, Thermo Scientific); p70 S6Kinase (H-9, #sc-8418, WB: 1:200), FASN (G-11, #sc-48357, IF: 1:200) (both Santa Cruz Biotechnology).

### Chemicals
Chemicals used were C75 (Sigma Aldrich), TOFA (5-(tetradecyloxy)-2-furoic acid), acetyl-CoA carboxylase inhibitor (Abcam, ab141578), Triascin C (Enzo, BML-EI218-100), Bromoenol Lactone (Cayman Chemical, 70700), Etomoxir (Calbiochem/MilliporeSigma), T-863 (Cayman Chemical, 25807), SB 204990 (Cayman Chemical, 15245), Cerulenin (Santa Cruz Biotechnology), GSK2194069 (Sigma Aldrich), CAY10566, GSK2606414 (both Cayman Chemical), Tunicamycin (Cayman Chemical), MMS (Santa Cruz Biotechnology), NCS (Sigma Aldrich), Torin2, Nutlin-3 (both Medchemexpress), DMSO (Sigma Aldrich). Concentrations used are noted in the Fig. legends and text.

### Reporting summary
Further information on research design is available in the Nature Portfolio Reporting Summary linked to this article.

## Data availability
The lipidomic data generated in this study is available in Supplementary Data 1. The RNA Sequencing data generated in this study have been deposited in the GEO (Gene Expression Omnibus) database under the accession code GSE254479. The differential gene expression data derived from the RNA Sequencing data are available in Supplementary Data 3. The lipid-mRNA correlations generated in this study based on the lipidomic and differential gene expression analysis are available in Supplementary Data 2. Databases used for data analysis include human genome assembly (GRCh38, https://www.ncbi.nlm.nih.gov/datasets/genome/GCF_000001405.26/), Gene Ontology (GO) by DAVID Bioinformatics (https://david.ncifcrf.gov/), ATF4-dependent genes taken from this publicly available RNA Sequencing dataset GSE158605. Uncropped scans of Western Blots are available in the Source Data file and in the Supplementary Information file. Raw images supporting the findings of this study are available from the corresponding authors upon reasonable request. Source data are provided with this paper.

## Code availability
The code detailing the fixed and live cell image analysis pipeline is available at https://github.com/scappell/Cell_tracking.

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

## Acknowledgements

We thank Hee Won Yang, Katie Ferrick, Leighton Daigh, Lindsey Pack, Lin Gan, and Nalin Ratnayeke for help with reagents and cell lines; Damien Garbett and Anjali Bisaria for imaging assistance; members of the Meyer and Jackson lab for helpful discussions and Tomek Swigut for feedback and technical support. We thank the Stanford Functional Genomics Facility for performing RNA-Sequencing and the Stanford High-Throughput Screening Knowledge Center (HTSKC) and D. E. Solow-Cordero for support and use of the ImageXpress Micro Confocal (SIG S10OD026899). We thank Lindsey Pack for review of the manuscript. We thank Janos Demeter for support with RNA sequencing data availability. This work was funded by a National Institute of General Medical Sciences grant R35GM127026. The lipidomics work was supported by the Mass Spectrometry Core of the Salk Institute with funding from NIH-NCI CCSG: P30 014195 and the Helmsley Center for Genomic Medicine. The MS data described here was gathered on a ThermoFisher Q Exactive Hybrid Quadrupole Orbitrap mass spectrometer funded by NIH grant (1S10OD021815-01). M.S.K. was supported by a postdoctoral fellowship from the Human Frontiers Science Program Organization. Y.F. was supported by the Stanford Graduate Fellowship and the Stanford Center for Systems Biology. C.L. was supported by the NSF Graduate Research Fellowship.

## Author contributions

M.S.K. and T.M. planned the study and designed the experiments. M.S.K. performed most of the experiments. Y.F. and C.L. helped in design, implementation, and computation of the RNA Sequencing experiment. Y.F., C.L., and M.C. helped with computation and tracking of live and fixed cell image analysis. A.F.M.P. and A.S. performed lipidome analysis. M.S.K., Y.F., C.L., M.C., A.F.M.P., P.K.J., A.S., T.M. analyzed and interpreted data. M.S.K and T.M. wrote the manuscript.

## Competing interests

The authors declare no competing interests.
