## [Peer Review File · Nature Communications]

A fast-acting lipid checkpoint in G1 prevents mitotic defectsREVIEWER COMMENTS

Reviewer #1 (Remarks to the Author):

Title: A fast-acting lipid checkpoint in G1 prevents cell death in mitosis

Authors: Marielle S. Köberlin, Yilin Fan, Chad Liu, Mingyu Chung, Antonio F. M. Pinto, Peter K. Jackson, Alan Saghatelian, & Tobias Meyer

Manuscript #: NCOMMS-23-33777

Summary: This is an impressive study that identifies changes both lipid metabolism and lipid composition combine to regulate G1/G0 transition to S-phase where appropriate lipid composition promotes survival. The work is important in that a role for lipids in cell cycle progression is under-appreciated. The only serious comment I have is the different representation of G1, G0 and late G1. It was reported previously that a lipid checkpoint was in late G1 before an mTOR-dependent checkpoint (Ref 40 in text, Patel et al.). It would be of interest to discuss how the lipid checkpoint(s) functions relative to G0, G1, and "late" G1. The authors could also clarify better what is the difference between G0 and G1 – is there an early G1 where cells are coming from mitosis and in the absence of growth factors (and lipids) cells exit to G0 where cells can be stimulated to re-enter the cell cycle – referred to as "EGF release". In Fig. 1A, there is a map that reads: G0 – G1 – S. In Fig. 2A, the map is: G1 – late G1 – S. In Fig. 2F, it is G0 – G1 – S (no late G1). In Fig. 5M, there is G1 – late G1 – S as in Fig. 2A. In Fig. 5M, there is some indication that the lipid checkpoint is in what they are calling "late G1", which is only mentioned in the text once. The authors do not make clear what constitutes "late G1". Although it is somewhat controversial, there might be some support for the Restriction Point of Arthur Pardee contained in the voluminous supplemental data provided.

Aside from the confusing descriptions of cell cycle progression, this is an important technical tour de force where many observations presented could lead to many follow-up papers on the significant roles that lipids play in cell cycle progression and survival.

Reviewer #2 (Remarks to the Author):

Koberlin et al. test the dependence of progression through the cell cycle on lipid synthesis. They mainly use a small molecular inhibitor against the fatty acid synthase FASN, which prevents non-transformed MCF10A cells from progressing through G1 in a dose dependent manner. They claim to identify a reversible cell cycle checkpoint that is activated downstream a UPR pathway triggered by low lipid synthesis in G1. This checkpoint arrests cells in G1 in the presence of the FASN inhibitor. They claim that this response is specific to non-transformed cells and test this in a different cell type that is also non-cancerous (RPE1).

The understanding of how cells regulate their lipid content throughout the cell cycle is an important and understudied area. The authors take a systems biology approach to determine the networks that control the cell cycle arrest in cells treated with the FASN inhibitor. They find that ER stress signaling and cell cycle regulators converge to arrest cells in G1. The outputs that result in cell cycle arrest are mostly convincing, but I have major concerns about the interpretation of the initial portions of the paper.

1. My major concern is that the authors overstate their claims that the "lipid checkpoint" results from low levels of fatty acid synthesis in G1 when most of their work is done with inhibition of FASN (with the C75 inhibitor). FASN inhibition increases the levels of malonyl-CoA, in some cells more than 8 fold, which inhibits mTORC directly (PMID: 37563253). Inhibition of mTOR arrests cells in the cell cycle.

Malonyl-CoA also inhibits fatty acid oxidation, which could have other side effects. Using TOFA to inhibit ACC because it does not result in the accumulation of malonyl-CoA would substantiate their conclusions.

2. The authors data show that addition of 5% serum restores the effects on the G1 arrest in cells treated with the FASN inhibitor, however because serum has many other factors beyond lipids, a more direct experiment would be to add fatty acids to test if fatty acids alone is sufficient to promote cell cycle progression in C75 treated cells. This experiment can be done with different types and ratios of fatty acids.

3. It appears as though the lipidomics is performed on a very small population of cells, so it is not clear how the authors are correlating their lipid

Other comments:

2. In general, the paper overstates claims making the paper difficult to follow. This can be remedied by taking out discussion points from the results sections and writing a more thorough and balanced introduction. Some examples are listed below:

-End of introduction pg. 3: "to prevent fatty acid insufficiency from triggering mitotic failure and cell death at the end of the cell cycle" How do we know that insufficiency triggers cell death. 14% of cells that enter mitosis have very aberrant mitoses, but there is no data in this paper that shows cell death pathways are "triggered". Could FASN inhibition, which also effects fatty acid breakdown impact mitochondrial and the cell death response? Cell death is also in their title and abstract.

-Introduction: However, cells must not only duplicate their DNA, but also their lipid mass during each cell cycle and also adapt the lipid composition for a successful mitosis⁹⁻¹². The papers cited here (9-12) strictly about cytokinesis. The authors should specifically replace "mitosis" with "cytokinesis". This should be done throughout the introduction, results and discussion in relation to citations 9-12.

-"Fatty acids that are synthesized late in the cell cycle are important for nuclear envelope formation⁴⁷, explaining why FASN inhibition in G2 can lead to mitotic failure. This sentence belongs in the discussion not in the results. There are other papers in which the lipid repertoire impacts mitotic progression that the authors did not include (PMID: 34852214, PMID: 30510774). Also the mitotic phenotypes reported in this paper are not nuclear envelope reformation defects – these cells appear to be highly defective in chromosome segregation or may be undergoing cell death.

- Top of page 6 the authors should write "suggest" not "show"

- page 7 the authors state "A shift from saturated to unsaturated lipid species increases membrane fluidity^{51,52}, which may be important for a successful completion of mitosis." Most cellular membranes are made up of unsaturated lipids. For example, the ER is the largest organelle, so much of the lipids they are analyzing are likely ER lipids. My expectation is that unsaturated lipids are synthesized to promote the increased size of the ER and other membranes not necessarily to increase membrane fluidity to complete cytokinesis (presuming the authors are referring to cytokinesis based on the prior citations 9-12).

- The authors state in the discussion that they expect that cancer cells have lost the G1 checkpoint, which is why they are sensitive to FASN inhibition. However, there are cancer cell types in culture that do not die from TOFA treatment. The authors should provide an explanation for why FASN inhibition would be different from TOFA inhibition.

Other comments:

1. The authors use siRNA and CRISPR-Cas9 gene editing to reduce/delete FASN but the difference they see in both cases are quite marginal (~10% difference). This is an important experiment to complement their FASN inhibitor results. For the RNAi-depletion, they claim that it is because the knockdown is not very good, but the antibody used in Fig. S1G is not listed and not tested in the FASN knockout cells in the same way. Also, it is unclear how they set up the experiment with the

knockdown/out of FASN? Figure S1f-H. Is this in an asynchronous population?

2. Figure 2 is confusing in the way that the data are described and also the figure is presented. The authors "analyze changes in the relative intracellular lipid abundance between starved, quiescent cells and those progressing through G1" To my understanding there is a very small population of cells progressing through G1 and entering Sphase at any one time point (based on the data in Figure 2B and also in Fig. 1). How does whole cell lipidomics of a population of cells pick up on differences occurring in this small population?

3. PC is the most abundant lipid in most human cells and makes up the majority of lipids in the ER (PMID: 18216768), yet the authors only note changes in PE, TG, PA and MG/DG. They state that "prominent ER lipids" change but the most prominent ER lipid is PC, which is not mentioned in this description. This makes this result confusing. A description of the different phospholipid classes and saturation would be more clear in an expanded introduction.

4. For figure 2F, the authors could use TOFA to inhibit ACC to substantiate their claims.

5. The coregulation analysis is very interesting and in my opinion is the strongest section of the manuscript. They say that their data suggested a critical role for the rapid induction of PERK upon lipid perturbation in G1. Might this also be true in other phases of the cell cycle? In Figure 1B they authors show a lot of cells in G2/M and this is increased in the FASN treated culture, but no cells in S-phase. Does this suggest that the arrest does not happen if there are cells in S-phase already?

6. They authors claim that FASN inhibition results in "lipid remodeling and new synthesis" (pg 16) is overstated. Remodeling may be upregulated so that cells can survive. The authors never measure synthesis to conclude that there is new synthesis over remodeling.

Minor:

1. On page 3 and 4, the authors claim that the inhibition of FASN was "acute" (for example, "We tested the importance of fatty acid synthesis by acutely blocking different catalytic active sites of the enzyme FASN". In the next paragraph they explain that they treated with FASN inhibitors for 20hours. Later, they state that FASN was "acutely" inhibited FASN when cells were treated with the inhibitor 9h after release. It is unclear what they define as acute? Also, there is no indication of how long they treated with the other inhibitors in the results or figure legend, which they also claim was "acute." (I was also unable to find it in the materials and methods.)

2. Figure 2C is group 3 missing?

3. Figure 4 there is not E?

4. In figure 1B, why not show higher concentrations? For example with TOFA? Is it because cells die?

5. It is difficult to follow paragraph in section entitled : "the lipidome is reshaped before S-phase begins and acute inhibition of FASN...". This should be rewritten to make sense of phase and type of lipids.

Reviewer #3 (Remarks to the Author):

Köberlin and colleagues present an interesting and robust manuscript documenting the existence of a lipid-dependent cell cycle arrest programme. They use a combination of lipidomics, transcriptomics and quantitative live cell imaging of cell cycle reporters to show that acute inhibition of Fatty Acid Synthase (FASN) triggers cell cycle arrest in G1. They show roles for the ER-stress response, induction of the CDK-inhibitor, p21 and reduction in levels of the key G1 cyclin, CyclinD1, in mediating this cell cycle arrest and propose the existence of a lipid-based cell cycle checkpoint.

I think the data are generally clear, robust and believable. I'm not an 'omics expert, but think that the

authors have done a good job to explain and integrate lipidomic and transcriptomic data. I followed most of it, except the gene cluster associations with specific lipids in Figure 4 and S4. Perhaps rewording for non-experts like this reviewer would help here. I thought that the description of early and late changing lipids across the cell cycle was really interesting and I was surprised at the dynamic changes in lipid classes as cells proceed through G1 and into S. The effects of FASN inhibition in 2G and 2H are also exciting and clear. There is one line in the abstract that states that they showed that synthesis of lipids in G1 increases the lipid mass – I couldn't see that data that measured mass.

The comments I have are mainly conceptual and related to data presentation.

1. The authors are keen to present this as a new 'checkpoint', rather than engagement of existent cell cycle control mechanisms. To me this brings with it a few problems that I think should be addressed:

a. The effects are induced by acute inhibition of FASN with a potent chemical inhibitor. If this is a checkpoint to sense lipid levels and commit to the cell cycle, under what physiological (or pathophysiological roles) is it invoked? Can the authors demonstrate that this arrest is engaged in response to physiological cues?

b. The effectors of this cell cycle arrest are suggested to be p21 and CLND1, and the ER-stress response is also suggested to contribute. That acute inhibition of FASN leads to ER-stress is perhaps unsurprising as you're stopping synthesis of new structural phospholipids. As the authors note on p13, the ER-stress response leads to cell cycle arrest in G1 by reducing CLND1 levels (refs 72&73), and possibly through controlling p21 (PMID 28975618). So, is the inhibition of FASN activity engaging a new cell cycle checkpoint, or is FASN activity simply elevating ER-stress and engaging already known mediators of cell cycle arrest in G1? Perhaps the authors could investigate the effects of FASN-inhibition in cells genetically incapable of mounting an ER-stress response?

c. For the relief of the cell cycle inhibition using serum lipids, I think this experiment is just returning the starved and EGF-released cells to complete medium, which does contain lipids in the serum, but also contains a lot of other things. Can the authors restore cycling by adding back defined lipids?

d. For canonical checkpoints acting in G1 (e.g., R), induction of the insult (e.g., GF withdrawal) after G1 leads to arrest in the next G1. The authors are showing that FASN inhibition after G1 leads to defects in mitosis, suggesting that cell cycle commitment is preserved. They analyse mitotic failure, but I was wondering what happened to cells that completed mitosis – do they then arrest in the next G1? Related to the separation of ER-stress-induced cell cycle arrest and the potential lipid checkpoint in G1, if you apply ER-stress in G1 or after G1, do you arrest in G1 or in the next G1? I think it would help to disentangle these possibilities.

e. I'm intrigued by the acuteness of the arrest. Presumably it takes some time to deplete existing fatty acids after FASN inhibition. Do you think the cell cycle arrest is mediated by a reduction in FFA mass, or by the activity of the synthase?

2. The data presentation is quite 'meta'. This is necessary for the lipidomics and transcriptomics approaches, and also for the output of the quantitative live cell imaging. However, some of the imaging-based expression data is not particularly convincing and not particularly accessible. I wonder if clearer protein level data (e.g., western blotting) can be provided for CLND1 and p21 levels in Figure 5A-D and S5? I find Log2 values quite hard to process, when a simple western would let readers see at a glance the strength of the authors' claims. For Figure 5B and S5B, if these are matched values can you provide the paring lines, and provide some sort of +ve control for levels of p21 induction after, say, DNA damage? The graphs for pRb phosphorylation are quite clear, but again, I'd like to see westerns of these levels.

Reviewer #1 (Remarks to the Author):

Title: A fast-acting lipid checkpoint in G1 prevents cell death in mitosis

Authors: Marielle S. Köberlin, Yilin Fan, Chad Liu, Mingyu Chung, Antonio F. M. Pinto, Peter K. Jackson, Alan Saghatelian, & Tobias Meyer

Manuscript #: NCOMMS-23-33777

Summary: This is an impressive study that identifies changes both lipid metabolism and lipid composition combine to regulate G1/G0 transition to S-phase where appropriate lipid composition promotes survival. The work is important in that a role for lipids in cell cycle progression is under-appreciated. The only serious comment I have is the different representation of G1, G0 and late G1. It was reported previously that a lipid checkpoint was in late G1 before an mTOR-dependent checkpoint (Ref 40 in text, Patel et al.). It would be of interest to discuss how the lipid checkpoint(s) functions relative to G0, G1, and “late” G1. The authors could also clarify better what is the difference between G0 and G1 – is there an early G1 where cells are coming from mitosis and in the absence of growth factors (and lipids) cells exit to G0 where cells can be stimulated to re-enter the cell cycle – referred to as “EGF release”.

We thank the reviewer for their thoughtful comments. Most of our experiments are performed in cells exiting quiescence (G0). We monitor the increase of Cyclin D transcription and translation (Fig. 5D E) which shows that there is signaling happening downstream of epidermal growth factor receptor (EGFR) that leads to S phase entry. However, in the presence of FASN inhibitors S phase is not entered so the cells stay in a G1 state (Fig. 1B, C). We have included the sentence in the text page 13: “Cyclin D levels are decreased compared to DMSO treated cells, however they are still increased compared to starved cells (Fig. 5E) indicating that the FASN inhibited cells have exited quiescence.” We have also included G0/G1 where applicable in the figures (Fig. 1B,C).

In Fig. 1A, there is a map that reads: G0 – G1 – S. In Fig. 2A, the map is: G1 – late G1 – S. In Fig. 2F, it is G0 – G1 – S (no late G1). In Fig. 5M, there is G1 – late G1 – S as in Fig. 2A. In Fig. 5M, there is some indication that the lipid checkpoint is in what they are calling “late G1”, which is only mentioned in the text once. The authors do not make clear what constitutes “late G1”. Although it is somewhat controversial, there might be some support for the Restriction Point of Arthur Pardee contained in the voluminous supplemental data provided.

We agree with the reviewer that we do not have sufficient markers to show exactly when the lipid checkpoint is engaged. From our data we can only conclude that inhibition of FASN between 9 and 12 hours after EGF release activates the lipid checkpoint signaling (Fig. 1E). We have unified the maps and are not referring to an early or late G1 in the revised text.

Aside from the confusing descriptions of cell cycle progression, this is an important

technical tour de force where many observations presented could lead to many follow-up papers on the significant roles that lipids play in cell cycle progression and survival.

We thank the reviewer for the positive comments and believe we clarified the description of the cell cycle progression.

Reviewer #2 (Remarks to the Author):

Koberlin et al. test the dependence of progression through the cell cycle on lipid synthesis. They mainly use a small molecular inhibitor against the fatty acid synthase FASN, which prevents non-transformed MCF10A cells from progressing through G1 in a dose dependent manner. They claim to identify a reversible cell cycle checkpoint that is activated downstream a UPR pathway triggered by low lipid synthesis in G1. This checkpoint arrests cells in G1 in the presence of the FASN inhibitor. They claim that this response is specific to non-transformed cells and test this in a different cell type that is also non-cancerous (RPE1).

The understanding of how cells regulate their lipid content throughout the cell cycle is an important and understudied area. The authors take a systems biology approach to determine the networks that control the cell cycle arrest in cells treated with the FASN inhibitor. They find that ER stress signaling and cell cycle regulators converge to arrest cells in G1. The outputs that result in cell cycle arrest are mostly convincing, but I have major concerns about the interpretation of the initial portions of the paper.

1. My major concern is that the authors overstate their claims that the “lipid checkpoint” results from low levels of fatty acid synthesis in G1 when most of their work is done with inhibition of FASN (with the C75 inhibitor). FASN inhibition increases the levels of malonyl-CoA, in some cells more than 8 fold, which inhibits mTORC directly (PMID: 37563253). Inhibition of mTOR arrests cells in the cell cycle. Malonyl-CoA also inhibits fatty acid oxidation, which could have other side effects. Using TOFA to inhibit ACC because it does not result in the accumulation of malonyl-CoA would substantiate their conclusions.

We thank the reviewer for raising this important point. To address the question of mTOR inhibition by FASN inhibition, we performed a western blot of phospho-p70S6K (T389) as a marker of mTOR signaling and we do not see a reduction in cells treated with C75 compared to the DMSO control after 4 hours or 20 hours (Fig. S6F). When we treat MCF-10A cells with Torin2, an mTOR activity inhibitor, we see a decrease in phospho-p70S6K and Cyclin D levels and no change in p21 levels (Fig. S6F, G), while treatment with different FASN inhibitors increases p21 (Fig. 5A-D). We have also included a western blot to show increased levels of p21 upon FASN inhibition (Fig. 5C). Treating MCF-10A cells with TOFA, as the reviewer suggested, also increases p21 and decreases Cyclin D (Fig. S7C), mimicking the treatment with FASN inhibitors. We also tested whether fatty acid oxidation was involved in the lipid checkpoint based on the

reviewer's suggestion but treatment with Etomoxir did not change p21 or Cyclin D levels (Fig. S6H).

We have included the following paragraph in the revised text: "To test whether mammalian Target of Rapamycin (mTOR) inhibition by increased levels of Malonyl-CoA was involved in the lipid checkpoint signaling (78), we measured phosphorylation levels of p70 S6 Kinase (p-T389) (79). FASN inhibition after 4 or 20 hours of EGF release did not inhibit mTOR signaling as compared to Torin2 treatment (Fig. S6F), a selective mTOR inhibitor (80). Further, Torin2 treatment during EGF release decreased Cyclin D levels and did not increase p21 levels after 4 hours, so it did not mimic FASN inhibition (Fig. 5C and S6G). To test if mitochondrial function was important for the lipid checkpoint, we inhibited fatty acid oxidation with Etomoxir or oxidative phosphorylation with Oligomycin (81). Both treatments did not increase p21 levels after 4 hours during EGF release (Fig. S6H, I). Etomoxir treatment did not decrease Cyclin D levels while Oligomycin did (Fig. S6H, I)."

Additionally, we show that when we inhibit fatty acid production via the phospholipase pathway using bromoenol lactone (BEL) to inhibit phospholipase activity we also see an increase in p21 and decrease in Cyclin D (Fig. S7A) further highlighting the importance of fatty acid production rather than inhibition of FASN and increased levels of Malonyl-CoA.

2. The authors data show that addition of 5% serum restores the effects on the G1 arrest in cells treated with the FASN inhibitor, however because serum has many other factors beyond lipids, a more direct experiment would be to add fatty acids to test if fatty acids alone is sufficient to promote cell cycle progression in C75 treated cells. This experiment can be done with different types and ratios of fatty acids.

As suggested by the reviewer we have added Palmitate complexed to fatty acid free BSA to MCF10A cells. These cells do not tolerate fatty acid free BSA very well so the overall levels of phospho-Rb positive cells are decreased in the DMSO condition. We further see a reduction of phospho-Rb levels in the presence of Palmitate, likely due to a stress sensitivity to the extra Palmitate. Regardless, in this rescue experiment we see an increase in phospho-Rb levels when we add Palmitate in the presence of the FASN inhibitor C75 (Fig. 5K). Even though cells show partial sensitivity to the BSA and Palmitate, this experiment shows that Palmitate addition can rescue FASN inhibition. We have added the following paragraph: "Additionally, supplementation of cells with Palmitate complexed to BSA rescued Rb (p-S807/S811) levels in the presence of FASN inhibition even though fatty-acid free BSA and Palmitate-BSA alone reduced the levels of Rb (pS807/811) positive cells (Fig. 5K). These data demonstrate that inhibition of FASN blocks S phase entry by disrupting Palmitate and lipid synthesis, and deregulating Rb signaling."

3. It appears as though the lipidomics is performed on a very small population of cells, so it is not clear how the authors are correlating their lipid

The time points for lipidome and transcriptome analysis of synchronized MCF-10A cells released out of quiescence were selected based on the percentage of change of phospho-Rb and EdU positive cells over time (Fig. S3A, B). After EGF release out of quiescence MCF10A cells exit G0 and progress through G1 to enter S phase. Since phospho-Rb precedes EdU incorporation, it is the first immunofluorescence read-out we can use to detect where in the cell cycle the cells are. This signal can be detected as early as 8h after EGF release. Between 16h and 20h post EGF release the amount of phospho-Rb positive cells has plateaued at around 55-60% as the first cells have already finished DNA replication and the last cells are just starting. Between 12 and 15 hours post EGF release most cells are right at the decision point of DNA replication or have just started. This is a narrow time window when the critical activity changes are happening, which motivated us to use the 12 and 15h timepoints after EGF release to perform the lipidome and transcriptome analysis.

Since the former Figure 2B was distracting from the actual focus of Figure 2, we have moved that panel to the supplementary data (Fig. S3A) and also added a detailed phospho-Rb time course analysis (Fig. S3B).

We have included the following sentence in the text: “Another marker for the G1/S transition is the phosphorylation of Retinoblastoma protein (Rb) (50–52).

Immunofluorescence analysis shows that up to 60% of cells are Rb (pS807/811) positive between 15-20 hours after EGF release indicating that most cells are in S phase at that point (Fig. S3B).”

Other comments:

2. In general, the paper overstates claims making the paper difficult to follow. This can be remedied by taking out discussion points from the results sections and writing a more thorough and balanced introduction. Some examples are listed below:

-End of introduction pg. 3: “to prevent fatty acid insufficiency from triggering mitotic failure and cell death at the end of the cell cycle” How do we know that insufficiency triggers cell death. 14% of cells that enter mitosis have very aberrant mitoses, but there is no data in this paper that shows cell death pathways are “triggered”. Could FASN inhibition, which also effects fatty acid breakdown impact mitochondrial and the cell death response? Cell death is also in their title and abstract.

We thank the reviewer for raising this point. We have removed cell death from the text, title and abstract and replaced it with mitotic defects.

In response to the other comment, we have treated MCF-10A cells with Oligomycin A and Etomoxir. Oligomycin A inhibits oxidative phosphorylation. In contrast to FASN inhibition, we see no change in p21 levels (Fig. S6I). Etomoxir inhibits fatty acid oxidation. Even at high doses of Etomoxir we do not see a change in pRb or EdU positive cells or p21 and Cyclin D levels (Fig. 1D, S1D and S6H). These data indicate that inhibition of oxidative phosphorylation leads to decreased levels of p21 and that fatty acid oxidation is not crucial for cell cycle progression/the lipid checkpoint.

We have added the following sentence in the revised text: “To test if mitochondrial function was important for the lipid checkpoint, we inhibited fatty acid oxidation with

Etomoxir or oxidative phosphorylation with Oligomycin (81). Both treatments did not increase p21 levels after 4 hours during EGF release (Fig. S6H, I). Etomoxir treatment did not decrease Cyclin D levels while Oligomycin did (Fig. S6H, I).”

-Introduction: However, cells must not only duplicate their DNA, but also their lipid mass during each cell cycle and also adapt the lipid composition for a successful mitosis^{9–12}. The papers cited here (9-12) strictly about cytokinesis. The authors should specifically replace “mitosis” with “cytokinesis”. This should be done throughout the introduction, results and discussion in relation to citations 9-12.

-“Fatty acids that are synthesized late in the cell cycle are important for nuclear envelope formation⁴⁷, explaining why FASN inhibition in G2 can lead to mitotic failure. This sentence belongs in the discussion not in the results. There are other papers in which the lipid repertoire impacts mitotic progression that the authors did not include (PMID: 34852214, PMID: 30510774). Also the mitotic phenotypes reported in this paper are not nuclear envelope reformation defects – these cells appear to be highly defective in chromosome segregation or may be undergoing cell death.

We agree that a body of excellent studies have been performed on the role of lipids which were shown to be particularly relevant in cytokinesis. The division of the nucleus also involves important lipid membrane rearrangements as highlighted by the references the reviewer suggests. There are also genome-wide studies showing that there is specific mitotic upregulation of lipid metabolism in mammalian and yeast cells (PMID: 28418768, PMID: 28057705). We have added these references and the reference (Scaglia, N. et al: De novo fatty acid synthesis at the mitotic exit is required to complete cellular division. *Cell Cycle* 13, 859–868 (2014).) to include both cytokinesis and mitosis in the citations in the introduction.

We also thank the reviewer for the other suggested references and have included them. And we have moved the suggested sentence to the discussion.

- Top of page 6 the authors should write “suggest” not “show”
We have changed the text according to the reviewer’s suggestion.

- page 7 the authors state “A shift from saturated to unsaturated lipid species increases membrane fluidity^{51,52}, which may be important for a successful completion of mitosis.” Most cellular membranes are made up of unsaturated lipids. For example, the ER is the largest organelle, so much of the lipids they are analyzing are likely ER lipids. My expectation is that unsaturated lipids are synthesized to promote the increased size of the ER and other membranes not necessarily to increase membrane fluidity to complete cytokinesis (presuming the authors are referring to cytokinesis based on the prior citations 9-12).

We have moved the sentence to the discussion and included the following: “...to promote the increased size of the ER during cell cycle progression.’

- The authors state in the discussion that they expect that cancer cells have lost the G1 checkpoint, which is why they are sensitive to FASN inhibition. However, there are cancer cell types in culture that do not die from TOFA treatment. The authors should provide an explanation for why FASN inhibition would be different from TOFA inhibition.

While some cancer cell types are sensitive to FASN inhibition, the majority of cancer cell lines are not FASN sensitive. Out of all the cancer cell lines analyzed by the Broad Institute on Depmap.org only 22% are sensitive to FASN inhibition, indicating that many cancer cell lines have adapted to not be dependent on de novo synthesis of fatty acids. Almost the same percentage of cancer cell lines are also dependent on ACACA (the enzyme targeted by TOFA). Additionally, there is a strong correlation between the FASN and ACACA sensitivity (see Figure 1 below). This means that most of the cancer cell types analyzed are both sensitive to FASN inhibition and ACACA inhibition.

Figure 1

Other comments:

1. The authors use siRNA and CRISPR-Cas9 gene editing to reduce/delete FASN but the difference they see in both cases are quite marginal (~10% difference). This is an important experiment to complement their FASN inhibitor results. For the RNAi-depletion, they claim that it is because the knockdown is not very good, but the antibody used in Fig. S1G is not listed and not tested in the FASN knockout cells in the same way. Also, it is unclear how they set up the experiment with the knockdown/out of FASN? Figure S1f-H. Is this in an asynchronous population?

We have added the antibody information and provided a detailed description of the FASN Crispr experiment to materials and methods.

2. Figure 2 is confusing in the way that the data are described and also the figure is presented. The authors “analyze changes in the relative intracellular lipid abundance between starved, quiescent cells and those progressing through G1” To my understanding there is a very small population of cells progressing through G1 and

entering Sphase at any one time point (based on the data in Figure 2B and also in Fig. 1). How does whole cell lipidomics of a population of cells pick up on differences occurring in this small population?

We addressed this in point (3) above and clarified in the revised text as follows: “Another marker for the G1/S transition is the phosphorylation of Retinoblastoma protein (Rb) (50–52). Immunofluorescence analysis shows that up to 60% of cells are Rb (pS807/811) positive between 15-20 hours after EGF release indicating that most cells are in S phase at that point (Fig. S3B).”

3. PC is the most abundant lipid in most human cells and makes up the majority of lipids in the ER (PMID: 18216768), yet the authors only note changes in PE, TG, PA and MG/DG. They state that “prominent ER lipids” change but the most prominent ER lipid is PC, which is not mentioned in this description. This makes this result confusing. A description of the different phospholipid classes and saturation would be more clear in an expanded introduction.

We have analyzed the enrichment of different lipid species in group 2 (increased lipids) and PC lipids are present (6th most abundant lipid species in that group) but not significantly enriched. That is why we have not mentioned PC species specifically. 7 out of the 12 PC species we measured are only moderately changed and cluster in group 3, so even though we see 5 PC species increase in group 2, it is not significantly enriched for PC.

We have now included a sentence in the introduction: “FASN synthesizes saturated fatty acids such as Palmitate, which can be desaturated and together are the building blocks of complex lipids including glycerophospholipids, sphingolipids, and neutral lipids.”

4. For figure 2F, the authors could use TOFA to inhibit ACC to substantiate their claims.

Since TOFA inhibition mimics FASN inhibition in many circumstances (Fig. 1D, S7C), we therefore predict that the lipidome changes would be very similar to FASN inhibition.

5. The coregulation analysis is very interesting and in my opinion is the strongest section of the manuscript. They say that their data suggested a critical role for the rapid induction of PERK upon lipid perturbation in G1. Might this also be true in other phases of the cell cycle? In Figure 1B they authors show a lot of cells in G2/M and this is increased in the FASN treated culture, but no cells in S-phase. Does this suggest that the arrest does not happen if there are cells in S-phase already?

We thank the reviewer for the positive feedback and for this suggestion. We performed a live cell experiment and gated for cells that are already in S phase when receiving the FASN inhibitor. Indeed, we do not observe an arrest in cells that are already in S phase (Fig. S2B). The same number of cells stay in a CDK high state as compared to DMSO. This is not the case for MMS treated cells for example.

We have added the following sentence in the revised text: “The population that was in S phase (CDK activity = 1-1.2) when treated with the FASN inhibitors showed no reduction in cells with high CDK activity as compared to cells treated with DNA damage inducing agent Methyl Methanesulfonate (MMS) (Fig. S2B)”.

We have included a quantification of cells in the different phases of the cell cycle in Figure 1C. There is a very marginal increase of 2% G2 cells in the C75 treated cell population.

6. They authors claim that FASN inhibition results in “lipid remodeling and new synthesis” (pg 16) is overstated. Remodeling may be upregulated so that cells can survive. The authors never measure synthesis to conclude that there is new synthesis over remodeling.

We agree with the reviewer and have removed “new synthesis” and replaced with “...resulting in the relative increase of lipid species including TG.”

Minor:

1. On page 3 and 4, the authors claim that the inhibition of FASN was “acute” (for example, “We tested the importance of fatty acid synthesis by acutely blocking different catalytic active sites of the enzyme FASN”. In the next paragraph they explain that they treated with FASN inhibitors for 20hours. Later, they state that FASN was “acutely” inhibited FASN when cells were treated with the inhibitor 9h after release. It is unclear what the define as acute? Also, there is no indication of how long they treated with the other inhibitors in the results or figure legend, which they also claim was “acute.” (I was also unable to find it in the materials and methods.)

We apologize for not being clearer on the timepoints and have added duration of treatment to the results, figure legends, and material and methods. We now only use “acute” for 3-hour treatments and have removed it for all other time points.

2. Figure 2C is group 3 missing?

We did not want to include these lipids, as to not distract form the other groups but we have now added the lipid numbers in Fig. S3D.

3. Figure 4 there is not E?

We apologize for this and have added Fig. 4E in the figure legend.

4. In figure 1B, why not show higher concentrations? For example with TOFA? Is it because cells die?

Yes, the cell number decreases. We titrated each compound used in this manuscript and only compare the concentrations in which we have comparable cell numbers with the control treatment. We have plotted the cell numbers at higher TOFA concentrations (see Figure 2 below).

Figure 2

5. It is difficult to follow paragraph in section entitled : “the lipidome is reshaped before S-phase begins and acute inhibition of FASN...”. This should be rewritten to make sense of phase and type of lipids.

We have re-written this paragraph to the following:

“To gain insights into how and when lipid changes in G1 control the lipid checkpoint, we analyzed the lipid composition as cells progress through G1 and enter S phase (Fig. 2A). S phase entry in synchronized MCF-10A cells starts between 12 to 15 hours post EGF release, as evidenced by incorporation of EdU (Fig. S3A). Another marker for the G1/S transition is the phosphorylation of Retinoblastoma protein (Rb) (50–52). Immunofluorescence analysis shows that up to 60% of cells are Rb (pS807/811) positive between 15-20 hours after EGF release indicating that most cells are in S phase at that point (Fig. S3B).

We utilized mass spectrometry-based lipidomics to analyze changes in the relative intracellular lipid abundance between starved, quiescent cells and those progressing through G1 (Fig. 2B). Markedly, rather than increasing all lipid species in G1, as one may expect from the need to double lipid mass at the time of cell division, there were strong, specific relative changes in lipid abundance before cells enter S phase (Fig. 2A, B and Table S1). By clustering the lipid changes we identified 4 groups of lipids.

Group 1 contained lipids that decreased with cell cycle progression and was 2- to 4-fold enriched for saturated Triglycerides (TG), Monoglycerides/Diglycerides (MG/DG),

Cholesterol/Cholesterol esters (Cho/ChE), and Phosphatidylserines (PS) (Fig. 2B, group 1, far left, and S3C, left).

Group 2 contained lipids that increased and was enriched up to 3-fold for glycerophospholipids including long-chained Phosphatidylethanolamines (PE), Phosphatidylinositols (PI), and most Phosphatidic acids (PA), as well as polyunsaturated TG (Fig. 2B, group 2, right, and S3C, right). PE lipids are important for membrane biosynthesis, and PI and PA mediate lipid signaling (53–55).

Group 3 contained lipids that did not change (Fig. 2B, center, and S3D), while Group 4 contained lipids that increased later in G1 and were enriched more than 7-fold for most unsaturated and some saturated free fatty acids (Fig. 2B, group 4, left, and S3C, center, S3E). These late increased lipids are likely not critical in G1 but become important later in the cell cycle.

The decreasing lipids in group 1 were significantly enriched for saturated lipids while the increasing lipids in group 2 were enriched for unsaturated lipids, and lipids in group 3 were not enriched for saturation levels (Fig. 2C). Together, our analysis identifies a decrease of saturated neutral lipids, as well as PS, and Cho/ChE. At the same time, there is an upregulation of unsaturated complex ER lipids PE, PI, PA, and TG. These lipids are important for membrane biosynthesis, fluidity, lipid signaling, and show the kinetic profile expected for lipid regulators needed to overcome a G1 lipid checkpoint response (Fig. 2D).

Reviewer #3 (Remarks to the Author):

Köberlin and colleagues present an interesting and robust manuscript documenting the existence of a lipid-dependent cell cycle arrest programme. They use a combination of lipidomics, transcriptomics and quantitative live cell imaging of cell cycle reporters to show that acute inhibition of Fatty Acid Synthase (FASN) triggers cell cycle arrest in G1. They show roles for the ER-stress response, induction of the CDK-inhibitor, p21 and reduction in levels of the key G1 cyclin, CyclinD1, in mediating this cell cycle arrest and propose the existence of a lipid-based cell cycle checkpoint.

I think the data are generally clear, robust and believable. I'm not an 'omics expert, but think that the authors have done a good job to explain and integrate lipidomic and transcriptomic data. I followed most of it, except the gene cluster associations with specific lipids in Figure 4 and S4. Perhaps rewording for non-experts like this reviewer would help here.

We thank the reviewer for their thoughtful comments and positive feedback. We have added and re-written the following sentences on page 10 and 11:

“We then calculated the correlation of each lipid abundance change with the changes in S phase entry for each condition and color-coded the nodes of the lipid network based on these correlation values (Fig. 4B).”

“We next extended the correlation analysis to link the lipid abundance changes to transcriptional changes. We identified all 3739 genes that were differentially regulated after three hours of FASN inhibition using C75 (Table S2). Then we calculated the correlation between the transcriptional changes and the lipid abundance changes for each of the 3739 genes across all perturbations. For each differentially regulated transcript we could color-code the lipid nodes on the network according to their correlation value.”

“We mapped the correlation of each differentially expressed gene with changes in lipid abundance on the network of 250 lipids and summarized the data in a heat map (Fig. 4C).”

I thought that the description of early and late changing lipids across the cell cycle was really interesting and I was surprised at the dynamic changes in lipid classes as cells proceed through G1 and into S. The effects of FASN inhibition in 2G and 2H are also exciting and clear. There is one line in the abstract that states that they showed that synthesis of lipids in G1 increases the lipid mass – I couldn't see that data that measured mass.

The introductory sentence about increasing lipid synthesis is based on prior work such as PMID: 913494 (cited in the introduction). We have re-written the sentence as following: “Lipid synthesis increases during the cell cycle to ensure sufficient membrane mass, but how insufficient synthesis restricts cell-cycle entry is not understood.”

The comments I have are mainly conceptual and related to data presentation.

1. The authors are keen to present this as a new 'checkpoint', rather than engagement of existent cell cycle control mechanisms. To me this brings with it a few problems that I think should be addressed:

a. The effects are induced by acute inhibition of FASN with a potent chemical inhibitor. If this is a checkpoint to sense lipid levels and commit to the cell cycle, under what

physiological (or pathophysiological roles) is it invoked? Can the authors demonstrate that this arrest is engaged in response to physiological cues?

Our serum starvation protocol we perform to synchronize MCF-10A cells uses the withdrawal of all extracellular lipids to arrest the cells in G0. In cancer cells that are acutely starved, p21 is upregulated to avoid cell death (PMID: 21887277). And since cancer cells have to adapt to different environmental conditions constantly including supply of lipids, cancer cells that have lost this lipid checkpoint have a survival advantage compared to cells that arrest when the lipid requirements are not ideal (PMID: 34766135).

b. The effectors of this cell cycle arrest are suggested to be p21 and CLND1, and the ER-stress response is also suggested to contribute. That acute inhibition of FASN leads to ER-stress is perhaps unsurprising as you're stopping synthesis of new structural phospholipids. As the authors note on p13, the ER-stress response leads to cell cycle arrest in G1 by reducing CLND1 levels (refs 72&73), and possibly through controlling p21 (PMID 28975618). So, is the inhibition of FASN activity engaging a new cell cycle checkpoint, or is FASN activity simply elevating ER-stress and engaging already known mediators of cell cycle arrest in G1? Perhaps the authors could investigate the effects of FASN-inhibition in cells genetically incapable of mounting an ER-stress response?

We thank the reviewer for these insightful and interesting questions. Since known ER stress responses are induced rapidly after FASN inhibition we hypothesize that the lipid stress is engaging already known mediators of cell cycle arrest (p21 and reduced levels of Cyclin D) but this is happening through a new checkpoint that is not dependent on DNA damage or p53. We have now included new data that shows that mTOR inhibition is also not involved in this lipid checkpoint (Fig. S6F, G). Our Tunicamycin treatment however shows that the ER stress induced by lipid changes is different from the ER stress induced by protein stress, since the lipid stress is engaged more rapidly (Fig. 5F) and is reversible (Fig. 5N) and Tunicamycin treatment is not.

Based on the feedback provided by the reviewer, we have treated cells with an inhibitor targeting PERK and could show that there is a partial rescue of pRb levels (Fig. S7F). We hypothesize that since there are several ER stress pathways others might be able to rescue the inhibition of PERK. We have included the following paragraph in the revised text: "To confirm that ER stress contributed to the decrease of Rb (p-S807/811) phosphorylation when FASN was inhibited, we used an inhibitor targeting PERK activation (Perki, GSK2606414) (82). Indeed, there was an increase in Rb (p-S807/811) positive cells when PERK was inhibited compared to cells treated with FASN inhibitor alone (Fig. S7F)."

c. For the relief of the cell cycle inhibition using serum lipids, I think this experiment is just returning the starved and EGF-released cells to complete medium, which does contain lipids in the serum, but also contains a lot of other things. Can the authors restore cycling by adding back defined lipids?

As suggested by the reviewer, we have added Palmitate complexed to fatty acid free BSA to MCF-10A cells. These cells do not tolerate fatty acid free BSA very well so the overall levels of phospho-Rb positive cells are decreased in the DMSO condition. We further see a reduction of phospho-Rb levels in the presence of Palmitate because the cells are sensitive to Palmitate levels as well. Regardless, we see an increase in phospho-Rb levels when we add Palmitate in the presence of the FASN inhibitor C75 (Fig. 5K). Because of the observations listed above we think that this experimental setup is not optimal, but it nevertheless shows that Palmitate alone is able to rescue FASN inhibition.

We have added the following paragraph: “Additionally, supplementation of cells with Palmitate complexed to BSA rescued Rb (p-S807/S811) levels in the presence of FASN inhibition even though fatty-acid free BSA and Palmitate-BSA alone reduced the levels of Rb (pS807/811) positive cells (Fig. 5K). These data demonstrate that inhibition of FASN blocks S phase entry by disrupting Palmitate and lipid synthesis, and deregulating Rb signaling.”

d. For canonical checkpoints acting in G1 (e.g., R), induction of the insult (e.g., GF withdrawal) after G1 leads to arrest in the next G1. The authors are showing that FASN inhibition after G1 leads to defects in mitosis, suggesting that cell cycle commitment is preserved. They analyse mitotic failure, but I was wondering what happened to cells that completed mitosis – do they then arrest in the next G1? Related to the separation of ER-stress-induced cell cycle arrest and the potential lipid checkpoint in G1, if you apply ER-stress in G1 or after G1, do you arrest in G1 or in the next G1? I think it would help to disentangle these possibilities.

We thank the reviewer for this suggestion. We treated asynchronously cycling MCF-10A cells with different FASN inhibitors and concentrations. Then we gated for cells that received the inhibitor 2h before mitosis. We calculated the percentage of cells that upregulated their CDK activity after mitosis (indicating S phase entry). We see a dose dependent arrest in the next G1 when FASN is inhibited (Fig, S2G, H).

So, our data shows that cells that are able to complete mitosis after FASN inhibition arrest in the next G1.

We added the following sentence in the revised text: “Daughter cells that successfully completed mitosis after treatment with FASN inhibitors arrested dose dependently in the following G1 (Fig. S2G, H).”

e. I'm intrigued by the acuteness of the arrest. Presumably it takes some time to deplete existing fatty acids after FASN inhibition. Do you think the cell cycle arrest is mediated by a reduction in FFA mass, or by the activity of the synthase?

Our lipidome analysis shows that many lipids change after 3 hours of FASN inhibition (Fig. 2F). This is our earliest time point we measured. This means it is possible to deplete fatty acids as soon as 3 hours (or earlier) from the cell. Additionally, we observe the same acute arrest and p21 induction, and Cyclin D reduction after inhibition of

phospholipase activity indicating that it is the reduction of FFA mass (Fig. S7A, B). We have now added data that shows that TOFA, the inhibition of ACACA (the enzyme that synthesizes Malonyl-CoA upstream of FASN) also mimics the effect of FASN inhibition (Fig. S7C). That is why we think the cell cycle arrest is mediated by the reduction of fatty acids and not the activity of the synthase.

We have added the following sentence in the revised text: “To show that the metabolic pathway upstream of FASN also impinges on p21 and Cyclin D, EGF-released cells were treated with the ACACA inhibitor TOFA. Indeed, p21 levels were increased and Cyclin D levels were decreased after 4 hours (Fig. S7C),”

2. The data presentation is quite ‘meta’. This is necessary for the lipidomics and transcriptomics approaches, and also for the output of the quantitative live cell imaging. However, some of the imaging-based expression data is not particularly convincing and not particularly accessible. I wonder if clearer protein level data (e.g., western blotting) can be provided for CLND1 and p21 levels in Figure 5A-D and S5? I find Log2 values quite hard to process, when a simple western would let readers see at a glance the strength of the authors’ claims.

We thank the reviewer for these suggestions. We have now added western blots for p21, Cyclin D and pRb (Fig. 5C and Fig, S6M). We have also added new microscopy images showing more cells (Fig. 5A)

For Figure 5B and S5B, if these are matched values can you provide the paring lines, and provide some sort of +ve control for levels of p21 induction after, say, DNA damage? The graphs for pRb phosphorylation are quite clear, but again, I’d like to see westerns of these levels.

We have added the paring lines for Fig. 5B and have added a positive control for p21 (Nutlin-3 treatment) (Fig. S6C, D).

We have added the following paragraph in the revised text: “As a positive control for p21 induction, we treated cells with Nutlin-3, a p53 activator that induces high p21 levels (77). Indeed, the increase of p21 levels using Nutlin-3 treatment was comparable to C75 treatment (Fig. S6C, D).”

REVIEWERS' COMMENTS

Reviewer #1 (Remarks to the Author):

The authors have addressed my concerns.

Reviewer #2 (Remarks to the Author):

The authors have addressed my concerns. One additional suggestion is for them to include in their abstract "in non-transformed cells" to make it clear that this checkpoint may not exist in transformed cancer cells. This addition will also help to eliminate confusion with prior work done in transformed cell lines.

Reviewer #3 (Remarks to the Author):

I'm happy with the author's revision. I think it's an exciting paper and worthy of publication.